# When and why microbial-explicit soil organic carbon models can be unstable

**Erik Schwarz**[1]**, Samia Ghersheen**[2]**, Salim Belyazid**[1]**, and Stefano Manzoni**[1]

[1]Department of Physical Geography and Bolin Centre for Climate Research, Stockholm University, Stockholm, Sweden
[2]Department of Soil and Environment, Swedish University of Agricultural Sciences, Uppsala, Sweden

**Correspondence:** Erik Schwarz (erik.schwarz@natgeo.su.se)

**Abstract.** `CE1`Microbial-explicit soil organic carbon (SOC) cycling models are increasingly being recognized for their advantages over linear models in describing SOC dynamics. These models are known to exhibit oscillations, but it is not clear when they yield stable vs. unstable equilibrium points (EPs) – i.e., EPs that exist analytically but are not stable in relation to small perturbations and cannot be reached by transient simulations. The occurrence of such unstable EPs can lead to unexpected model behavior in transient simulations or unrealistic predictions of steady-state soil organic carbon (SOC) stocks. Here, we ask when and why unstable EPs can occur in an archetypal microbial-explicit model (representing SOC, dissolved OC (DOC), microbial biomass, and extracellular enzymes) and some simplified versions of it. Further, if a model formulation allows for physically meaningful but unstable EPs, can we find constraints in the model parameters (i.e., environmental conditions and microbial traits) that ensure stability of the EPs? We use analytical, numerical, and descriptive tools to answer these questions. We found that instability can occur when the resupply of a growth substrate (DOC) is (via a positive feedback loop) dependent on its abundance. We identified a conservative, sufficient condition in terms of model parameters to ensure the stability of EPs. Principally, three distinct strategies can avoid instability: (1) negligence of explicit DOC dynamics, (2) biomass-independent uptake rate, or (3) correlation between parameter values to obey the stability criterion. While the first two approaches simplify some mechanistic processes, the third approach points to the interactive effects of environmental conditions and parameters describing microbial physiology, highlighting the relevance of basic ecological principles for the avoidance of unrealistic (i.e., unstable) simulation outcomes. These insights can help to improve the applicability of microbial-explicit models, aid our understanding of the dynamics of these models, and highlight the relation between mathematical requirements and (in silico) microbial ecology.

## 1 Introduction

Current Earth system models (ESMs) have very simplified representations of soil organic carbon (SOC) dynamics (Bradford et al., 2016; Todd-Brown et al., 2013; Varney et al., 2022). Accuracy in matching observed SOC stocks and turnover times has not significantly improved in the latest ensemble of ESMs used in the Coupled Model Intercomparison Project (CMIP, CMIP6) (Varney et al., 2022), with uncertainty about SOC responses to climate change remaining high (Todd-Brown et al., 2013; Varney et al., 2022). Consequently, a need to improve and diversify the description of SOC dynamics in ESMs has been identified (Bradford et al., 2016; Todd-Brown et al., 2013; Varney et al., 2022; Wieder et al., 2015, 2018). Current ESMs employ linear degradation kinetics to simulate SOC degradation (Todd-Brown et al., 2013). Thereby, they miss to integrate important aspects of our current understanding of major controls on SOC fate and to acknowledge the uncertainties in describing these processes (Abramoff et al., 2018; Abs et al., 2023; Bradford et al., 2016; Wieder et al., 2015, 2018). Non-linear, microbial-explicit SOC models can improve model–data agreement (Hararuk et al., 2015; Wieder et al., 2013). These models vary in terms of the number and identity of C pools and the degree of non-linearity (e.g., Allison et al., 2010; Manzoni and Porporato, 2009;

Schimel and Weintraub, 2003; Wang et al., 2013, 2015; Wieder et al., 2014, 2015). Among these models, the AWB (Allison–Wallenstein–Bradford) model (Allison et al., 2010) has emerged as an archetypal model structure to study the in-fluence of soil microbial processes on carbon stocks (e.g., Abs et al., 2022; Calabrese et al., 2022; Georgiou et al., 2017; Hararuk et al., 2015; Tao et al., 2023; Wieder et al., 2015). The AWB model explicitly represents pools of micro-bial biomass, extracellular enzymes produced by microbes, polymeric SOC that is not available for microbial uptake, and a pool of available dissolved organic carbon (DOC) produced from enzymatic depolymerization of SOC (Allison et al., 2010, Fig. 1a). With only four C pools and, commonly, two non-linear terms, the AWB model retains a comparably sim-ple structure and remains somewhat analytically tractable.

While better at predicting modern-day SOC stocks, microbial-explicit SOC models are known to exhibit oscil-latory behavior (e.g., Georgiou et al., 2017; Manzoni and Porporato, 2007; Sierra and Müller, 2015; Wang et al., 2014, 2016). Such oscillations can represent carbon–microbe dynamics observed at small spatial scales (see, e.g., discus-sion in Manzoni and Porporato, 2007) but are unfavorable for application at larger spatial and temporal scales, where such oscillations are generally not observed (Georgiou et al., 2017; Wang et al., 2014). A few studies have analyzed the os-cillatory properties of some microbial-explicit SOC models (Georgiou et al., 2017; Manzoni and Porporato, 2007; Wang et al., 2014, 2016). These studies characterized the dynam-ics exhibited after a perturbation around a model's equilib-rium (i.e., when all state variables are at steady state): does a model directly converge back to its previous equilibrium or does it approach the equilibrium with dampened oscil-lations? Different degrees of oscillatory behavior have been described, but, generally, these models were found to be sta-ble (that is, they do converge back to their previous equi-librium) for given parameterizations or if they follow ba-sic principles such as mass conservation and dependence of fluxes on source pools (Sierra and Müller, 2015; Wang et al., 2014, 2016). Stable oscillatory behavior, however, is only one of the possible dynamics such non-linear models can ex-hibit. In fact, these models can also be unstable (that is, after perturbation, a model does not converge back to its previous equilibrium) (Abs et al., 2022; Raupach, 2007; Schimel and Weintraub, 2003; Sierra and Müller, 2015), but the occur-rence of unstable equilibria in microbial-explicit SOC mod-els remains largely unexplored. While unstable equilibrium points exist analytically, they can never be reached by tran-sient simulations. Thus, model parameterizations that yield unstable equilibria can lead to unpredictable simulation out-comes as amplifying oscillations can occur; expected equi-librium states may not be reached (because they are unsta-ble), hindering convergence in model spin-up; or (some) state variables might collapse (e.g., Fig. 1b, yellow line). Further, if C stocks are predicted based on analytical steady-state so-lutions, unstable equilibria might lead to unrealistic predic-tions, causing mismatched outcomes from dynamic simula-tions. To increase the reliability of model predictions and model applicability, it is important to understand when and for what reasons microbial-explicit SOC models become un-stable.

Here, we study an archetypal microbial-explicit SOC model (based on the AWB model of Allison et al., 2010, and some simplified versions of it) to answer the follow-ing questions: (1) what mechanisms in microbial-explicit SOC models (model structures, used kinetic formulations, and parameter values) cause unstable equilibrium points to emerge? (2) How can we select model structures and/or con-strain model parameters to ensure the stability of equilibrium points?

## 2  Methods

### 2.1  Archetypal microbial-explicit SOC model

We start by defining the C mass balance equations for a sys-tem encompassing SOC ($S$), dissolved organic C (DOC, $D$), microbial biomass C (MBC, $B$), and extracellular enzyme C (ENZ, $E$) (Eqs. 1–4). The C compartments and flows are il-lustrated in Fig. 1a, and symbols for the variables and fluxes are defined in Tables 1 and 3. The C mass balance equa-tions are written as a system of ordinary differential equa-tions (ODEs), where, for convenience, the fluxes are aligned vertically according to their meaning (from left to right: ex-ternal inputs, depolymerization, uptake and metabolism, de-cay, and finally abiotic losses).

$$\frac{dS}{dt} = f_I I - P + f_D r_B D_B - L_S \tag{1}$$

$$\frac{dD}{dt} = (1 - f_I)I + P - U + (1 - f_D)r_B D_B + D_E - L_D \tag{2}$$

$$\frac{dB}{dt} = y_B U - R_E - D_B \tag{3}$$

$$\frac{dE}{dt} = (y_m - y_B)U + R_E - D_E - L_E \tag{4}$$

Organic matter enters the system with flux $I$, which is par-titioned between SOC and DOC depending on the fraction $f_I$. SOC is depolymerized by extracellular enzymes at rate $P$ and then is transferred to the DOC pool. DOC is directly available for microbial uptake at rate $U$. Both $P$ and $U$ are non-linear functions that can take on various forms (Table 2).

Microbes assimilate the substrate with a maximal effi-ciency $y_m \leq 1$ that is limited by physiological and/or thermo-dynamic constraints (Chakrawal et al., 2022) and use the sub-strate either for growth (i.e., biomass production at rate $y_B U$) or to produce extracellular enzymes (at rate $(y_m - y_B)U$). We refer to this uptake-dependent pathway of extracellu-lar enzyme production as "inducible" ENZ production. An alternative or complementary mode of ENZ production is

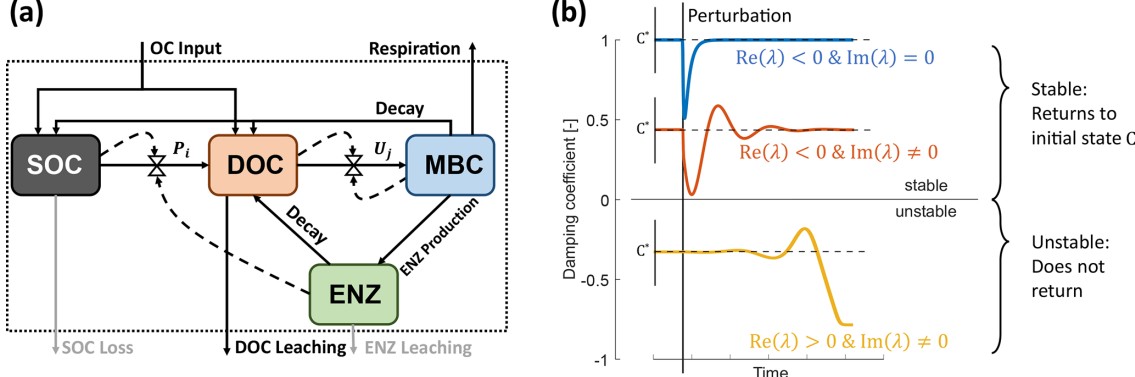

**Figure 1.** TS1 Model schematics of the archetypal microbial-explicit SOC model **(a)** and its relevant stability behaviors **(b)**. Colored boxes in **(a)** indicate the state variables of soil organic carbon (SOC), dissolved organic carbon (DOC), microbial biomass carbon (MBC), and extracellular enzymes (ENZs). Solid arrows indicate carbon fluxes, and dashed arrows connected to valve symbols indicate controls over the non-linear kinetic. Gray arrows indicate processes neglected in some analyses. The dotted box delineates the system's boundary. Colored lines in **(b)** illustrate the dynamics of a state variable relative to its steady-state value ($C^*$) following a perturbation for a stable node (damping coefficient $= 1$; blue line), a stable focus ($0 <$ damping coefficient $< 1$; red line), and an unstable focus ($-1 <$ damping coefficient $< 0$; yellow line). The subplot axes are centered around the value of their respective damping coefficient.

the biomass-dependent "constitutive" ENZ production at rate $R_E$, given by

$$R_E = m_E B, \tag{5}$$

where $m_E$ is the rate constant of constitutive ENZ production. In both formulations, we assumed that respiratory costs associated with enzyme production are already included in the growth respiration (proportional to $1 - y_m$). Two limiting cases can be derived from this general description of extracellular enzyme production:

Only constitutive ENZ production: $\quad y_B = y_m. \tag{6}$

Only inducible ENZ production: $\quad m_E = 0. \tag{7}$

Both MBC and ENZ are assumed to decay with a linear decay rate $D_i$:

$$D_i = d_i\, i, \quad (i = B, E), \tag{8}$$

but we also consider density-dependent microbial decay to be an alternative to the linear kinetic ($D'_B = d'_B B^b$, with $1 < b \le 2$; Georgiou et al., 2017). All decayed ENZs are assumed to return to the DOC pool, while only a fraction $r_B$ of the decayed microbial biomass is recycled in the system and partitioned between SOC and DOC according to the factor $f_D$. In turn, $(1 - r_B)D_B$ represents linear microbial maintenance respiration. SOC, DOC, and ENZs can have abiotic losses $L_i$ (e.g., erosion, leaching).

$$L_i = l_i\, i, \quad (i = S, D, E) \tag{9}$$

The system of Eqs. (1)–(4) constitutes a model of SOC cycling of varying complexity depending on the chosen kinetics. We refer to this four-pool model version as the SDBE

model according to the represented state variables. We use this system as a starting point for our analysis but reduce it to simpler variants to derive specific analytical results (Sect. 2.2).

Commonly, depolymerization of SOC by extracellular enzymes is described by either multiplicative ($m$), forward Michaelis–Menten ($f$), reverse Michaelis–Menten ($r$) (Schimel and Weintraub, 2003), or equilibrium chemistry approximation (ECA, $e$) (Tang and Riley, 2013) kinetics. Table 2 lists the respective formulations of $P_i$ ($i = m, f, r, e$), where $v_i^p$ is the maximal depolymerization rate coefficient, and $K_i^p$ is the respective half-saturation constant (if applicable). The uptake of DOC by microbes can be described with similar formulations, such as $U_j$ ($j = m, f$), simply by replacing $S$ with $D$ and $E$ with $B$ (Table 2), where $v_j^u$ is the maximal uptake rate coefficient, and $K_f^u$ is the respective half-saturation constant.

Many combinations of depolymerization and uptake kinetics are possible. For model versions with both non-linear terms, we limit our analysis to only a few combinations of depolymerization and uptake kinetics (indicated by the subscript $i \times j$ for the $i$th depolymerization kinetic and $j$th uptake kinetic), namely $m \times m$, $f \times f$, and $r \times f$ (see the summary of analyzed scenarios in Table 4). The first combination employing only multiplicative kinetics facilitates analytical tractability, the second combination is commonly used in other models (e.g., Allison et al., 2010; Georgiou et al., 2017; Tao et al., 2023), and the third combination is based on the conclusions of Tang and Riley (2019) that $r \times f$ might be an appropriate (and analytically tractable) approximation of ECA kinetics.

To improve the analytical tractability of the four-pool model, we neglect abiotic losses of SOC and ENZ by setting

**Table 1.** Description of all state variables and fluxes.

| Symbol | Description | Unit |
|---|---|---|
| *State variables* | | |
| $S$ | Soil organic carbon (SOC) | $\mathrm{mg\,C\,g^{-1}}$ |
| $D$ | Dissolved organic carbon (DOC) | $\mathrm{mg\,C\,g^{-1}}$ |
| $B$ | Microbial biomass (MBC) | $\mathrm{mg\,C\,g^{-1}}$ |
| $E$ | Extracellular enzymes (ENZs) | $\mathrm{mg\,C\,g^{-1}}$ |
| *Fluxes* | | |
| $I$ | Organic carbon input | $\mathrm{mg\,C\,g^{-1}\,d^{-1}}$ |
| $P$ | Depolymerization of SOC | $\mathrm{mg\,C\,g^{-1}\,d^{-1}}$ |
| $U$ | Microbial uptake of DOC | $\mathrm{mg\,C\,g^{-1}\,d^{-1}}$ |
| $L_S$ | Loss of SOC | $\mathrm{mg\,C\,g^{-1}\,d^{-1}}$ |
| $L_D$ | Leaching of DOC | $\mathrm{mg\,C\,g^{-1}\,d^{-1}}$ |
| $L_E$ | Leaching of extracellular enzymes | $\mathrm{mg\,C\,g^{-1}\,d^{-1}}$ |
| $D_B$ | Decay of microbial biomass | $\mathrm{mg\,C\,g^{-1}\,d^{-1}}$ |
| $D_E$ | Decay of extracellular enzymes | $\mathrm{mg\,C\,g^{-1}\,d^{-1}}$ |
| $R_E$ | Constitutive production of extracellular enzymes | $\mathrm{mg\,C\,g^{-1}\,d^{-1}}$ |

**Table 2.** Employed non-linear kinetics for the depolymerization rate $P_i$ and uptake rate $U_j$. Note that $m$ indicates multiplicative, $f$ indicates forward Michaelis–Menten, $r$ indicates reverse Michaelis–Menten (Schimel and Weintraub, 2003), and $e$ indicates equilibrium chemistry approximation (Tang and Riley, 2013). $U_r$ and $U_e$ are not used in our analysis.

| Kinetic $(i\,|\,j)$ | $P_i$ | $U_j$ |
|---|---|---|
| $m$ | $v_m^p S E$ | $v_m^u D B$ |
| $f$ | $v_f^p \dfrac{S}{K_f^p + S} E$ | $v_f^u \dfrac{D}{K_f^u + D} B$ |
| $r$ | $v_r^p S \dfrac{E}{K_r^p + E}$ | – |
| $e$ | $v_e^p \dfrac{S E}{K_e^p + S + E}$ | – |

$L_S = L_E = 0$ and by limiting the analysis only to the case of constitutive ENZ production ($y_B = y_m$, Eq. 6) (Table 4). The Jacobian matrix of partial derivatives for the four-pool model $J_{i \times j}^{\mathrm{SDBE}}$ is given in Eq. (A17). The matrix $J_{i \times j}^{\mathrm{SDBE}}$ is expressed in terms of general depolymerization and uptake kinetics $P_i$ and $U_j$, which allows for stability analysis of the SDBE model irrespective of the specific kinetics.

## 2.2 Reduced models for mathematical analysis

To identify which and how structural elements of the four-pool model with two non-linear kinetics affect model stability, we introduce two reduced model versions:

1. the SBE (SOC–MBC–ENZ) model, neglecting DOC dynamics, and

2. the SDB (SOC–DOC–MBC) model, assuming ENZ to be at a quasi-steady state.

Both model versions have only three pools but are different as, in the SBE model, only one non-linear term remains, while the SDB model still has both non-linear depolymerization and uptake kinetics. We analyze the former, less non-linear model, for all depolymerization kinetics listed in Table 2 with both constitutive and inducible ENZ production pathways and including abiotic losses of $S$ and $E$ (Table 4). In contrast, we analyze the latter, more non-linear model, after applying the same simplifying assumptions as for the four-pool model (Table 4) – that is, we set $L_S = L_E = 0$, $y_B = y_m$, and we only consider three combinations of depolymerization and uptake kinetics ($m \times m$, $f \times f$, and $r \times f$).

### 2.2.1 SBE model

DOC dynamics are neglected in the SBE (SOC–MBC–ENZ) model. Instead, it is assumed that any organic carbon that is made available by depolymerization of SOC is directly taken up by microbes – that is, $U = P$. The flux of decayed extracellular enzymes $D_E$ enters the SOC pool, and the partitioning factors $f_I$ and $f_D$ are set to 1. The resulting system of equations is given for the $i$th kinetic formulation for $P$ (Table 2) by

$$\frac{\mathrm{d}S}{\mathrm{d}t} = I - P_i + r_B D_B + D_E - L_S, \tag{10}$$

$$\frac{\mathrm{d}B}{\mathrm{d}t} = y_B P_i - R_E - D_B, \tag{11}$$

$$\frac{\mathrm{d}E}{\mathrm{d}t} = (y_m - y_B) P_i + R_E - D_E - L_E. \tag{12}$$

Note that, in this formulation, unless ENZ production is purely constitutive (Eq. 6), ENZ production is (partly) independent of microbial biomass (as $P_i$ only depends on $E$

and not on $B$). Consequently, as soon as there are extracellular enzymes that catalyze depolymerization, further enzyme production follows. The Jacobian matrix of the SBE model $J_i^{\text{SBE}}$ is a $3 \times 3$ matrix given by Eq. (A1).

### 2.2.2 SDB model

In the SDB (SOC–DOC–MBC) model, the extracellular enzyme pool is assumed to be at a quasi-steady state; that is, $\frac{\mathrm{d}E}{\mathrm{d}t} = 0$. With $L_E = 0$ and $y_B = y_m$, we obtain the quasi-steady-state concentration of $E$ from Eqs. (4), (5), and (8) as

$$E^{\text{qss}} = \frac{m_E}{d_E} B. \tag{13}$$

The SDB model is obtained by substituting $E^{\text{qss}}$ for $E$ in Eqs. (1)–(3), which yields the following after assuming $L_S = 0$:

$$\frac{\mathrm{d}S}{\mathrm{d}t} = f_I I - P_i^{\text{qss}} \qquad\qquad + f_D r_B D_B, \tag{14}$$

$$\frac{\mathrm{d}D}{\mathrm{d}t} = (1 - f_I)I + P_i^{\text{qss}} - U_j + (1 - f_D)r_B$$
$$D_B + D_E^{\text{qss}} - L_D, \tag{15}$$

$$\frac{\mathrm{d}B}{\mathrm{d}t} = y_m U_j - R_E - D_B. \tag{16}$$

In this model version, two non-linearities remain. Substituting $E^{\text{qss}}$ for $E$ in $D_E$ and $P_i$, we obtain $D_E^{\text{qss}} = R_E$ and $P_i^{\text{qss}} = f(S, B)$, respectively. The Jacobian matrix of the SDB model, $J_{i \times j}^{\text{SDB}}$, is a $3 \times 3$ matrix given by Eq. (A10).

### 2.3 Stability analysis

Here, stability behavior refers to how a model responds to a small perturbation around an equilibrium point (illustrated in Fig. 1b). The equilibrium points are obtained by assuming that the system behavior does not change over time – i.e., by setting the ODEs of all state variables $C$ to be equal to zero ($\frac{\mathrm{d}C}{\mathrm{d}t} = 0$). This yields their steady states $C^*$. For non-linear systems, stability is determined by the signs of the eigenvalues ($\lambda$) of the Jacobian matrix $J$ evaluated at an equilibrium point (denoted as $J\big|_*$; $\text{eig}(J\big|_*) = \lambda$) (e.g., Argyris et al., 2015). Briefly, if the real parts of all $\lambda$ eigenvalues are negative ($\text{Re}(\lambda) < 0$), the equilibrium point is stable: the system will converge back to this equilibrium point after a perturbation. Instead, if one or more eigenvalues have positive real parts ($\text{Re}(\lambda) > 0$), the equilibrium point is unstable, and the system will not return to the same state. If the eigenvalues additionally have non-zero imaginary parts ($\text{Im}(\lambda) \neq 0$), oscillations around the equilibrium point occur (Fig. 1b). Stability analysis is described in more detail in, e.g., Argyris et al. (2015).

### 2.3.1 Analytical approach

Because the eigenvalues of the Jacobian matrix can be analytically cumbersome, even in the comparably compact three-pool models, we use the Routh–Hurwitz criterion (e.g., Argyris et al., 2015; Horn and Johnson, 1994) for $J\big|_*$. The Routh–Hurwitz criterion states that all $\text{Re}(\lambda)$ vales have negative signs if, and only if,

1. all coefficients $a_i$ of the characteristic polynomial $\det(J\big|_* - \mathbf{1}\lambda) = 0$ (where $\mathbf{1}$ is the identity matrix; Eqs. 17 and 18) are positive (i.e., $a_i > 0$), and

2. $a_1 a_2 - a_3 > 0$ (if $J\big|_*$ is a $3 \times 3$ matrix) or $a_1 a_2 a_3 - a_3^2 - a_1^2 a_4 > 0$ (if $J\big|_*$ is a $4 \times 4$ matrix).

Thus, by applying the Routh–Hurwitz criterion we can analytically evaluate the stability around the equilibrium points of the non-linear systems given by the three- and four-pool models without directly evaluating $\lambda$ analytically. The characteristic polynomial for a $3 \times 3$ matrix is given by

$$\lambda^3 + a_1\lambda^2 + a_2\lambda + a_3 = 0, \tag{17}$$

and for a $4 \times 4$ matrix, it is given by

$$\lambda^4 + a_1\lambda^3 + a_2\lambda^2 + a_3\lambda + a_4 = 0. \tag{18}$$

In both cases, $a_1$ is the negative trace of $J\big|_*$ ($a_1 = -\text{tr}(J\big|_*)$).

### 2.3.2 Numerical simulations

We also compute $\lambda$ and the steady-state values of the state variables numerically. If not otherwise specified, $100\,000$ Monte Carlo simulations were produced by randomly drawing parameter values from (log-)uniform distributions using a Latin hypercube sampling algorithm (MATLAB R2022b's lhsdesign function; The MathWorks Inc., 2022). All parameters and their respective ranges are listed in Table 3. Partitioning coefficients were sampled from uniform distributions, while rate constants were $\log_{10}$-transformed before sampling.

Following Georgiou et al. (2017) and Sierra and Müller (2015), the stability of equilibria was evaluated using the damping coefficient given by

$$\zeta = \min\left[\frac{-\text{Re}(\lambda)}{\sqrt{\text{Re}(\lambda)^2 + \text{Im}(\lambda)^2}}\right], \tag{19}$$

which ranges between $-1$ and $1$. Note that $\zeta$ has positive values only if all $\text{Re}(\lambda) < 0$, indicating a stable equilibrium point, and it has negative values if any $\text{Re}(\lambda) > 0$, indicating an unstable equilibrium point. For $\text{Im}(\lambda) = 0$, $\zeta$ is either $1$ or $-1$, indicating no oscillations, while $-1 < \zeta < 1$ for $\text{Im}(\lambda) \neq 0$ indicates that oscillations occur.

Numerical simulations were carried out in MATLAB 2022b (The MathWorks Inc., 2022), and color maps created by (Crameri, 2018a, b) were used for visualization.

### 2.3.3 Classification of equilibrium points

Our analyses only consider physically meaningful equilibrium points – that is, only equilibrium points for which all

**Table 3.** Description of all parameters, their units, and used ranges for Monte Carlo simulations. Where applicable, parameterizations of limiting cases are separated by |. Parameter ranges where derived from Hararuk et al. (2015); Tao et al. (2023) and Cotrufo and Lavallee (2022). Baseline values were based on "conventional" values defined by Tao et al. (2023). See Sect. S1 in the Supplement for the derivation of parameter ranges for $m$, $r$, and $l$ kinetics, as well as for $d'_B$ and $I$.

| Symbol | Description | Unit | Baseline | Range |
|---|---|---|---|---|
| **Rate constants** | | | | |
| $I$ | Organic C input rate | $\mathrm{mg\,C\,g^{-1}\,d^{-1}}$ | varied | $1.88 \times 10^{-4} - 2.43 \times 10^{-2}$ |
| $v_m^p$ | Depolymerization rate coefficient ($m$) | $\mathrm{g\,mg\,C^{-1}\,d^{-1}}$ | $1.99 \times 10^{-1}$ | $9.13 \times 10^{-3} - 5.48 \times 10^{3}$ |
| $v_f^p$ | Depolymerization rate coefficient ($f$) | $\mathrm{d^{-1}}$ | TS2 $5.93 \times 10^{1}$ | $9.13 - 2.74 \times 10^{5}$ |
| $v_r^p$ | Depolymerization rate coefficient ($r$) | $\mathrm{d^{-1}}$ | $2.49 \times 10^{-1}$ | $9.13 \times 10^{-2} - 2.74$ |
| $K_f^p$ | Depolymerization half-saturation constant ($f$) | $\mathrm{mg\,C\,g^{-1}}$ | $3.00 \times 10^{2}$ | $5.00 \times 10^{1} - 1.00 \times 10^{3}$ |
| $K_r^p$ | Depolymerization half-saturation constant ($r$) | $\mathrm{mg\,C\,g^{-1}}$ | $2.00 \times 10^{-1}$ | $2.50 \times 10^{-2} - 3.00$ |
| $v_m^u$ | Uptake rate coefficient ($m$) | $\mathrm{g\,mg\,C^{-1}\,d^{-1}}$ | $1.25$ | $3.04 \times 10^{-2} - 1.10 \times 10^{2}$ |
| $v_f^u$ | Uptake rate coefficient ($f$) | $\mathrm{d^{-1}}$ | $2.49 \times 10^{-1}$ | $9.13 \times 10^{-2} - 2.74$ |
| $v_l^u$ | Linear uptake rate coefficient | $\mathrm{d^{-1}}$ | $1.25$ | $3.04 \times 10^{-3} - 1.10 \times 10^{1}$ |
| $K_f^u$ | Uptake half-saturation constant ($f$) | $\mathrm{mg\,C\,g^{-1}}$ | $2.00 \times 10^{-1}$ | $2.50 \times 10^{-2} - 3.00$ |
| $l_S$ | Loss rate coefficient of SOC | $\mathrm{d^{-1}}$ | $0$ | – |
| $l_D$ | Leaching rate coefficient of DOC | $\mathrm{d^{-1}}$ | varied $\mid 0$ | $2.74 \times 10^{-4} - 2.74 \times 10^{-1} \mid 0$ |
| $l_E$ | Leaching rate coefficient of extracellular enzymes | $\mathrm{d^{-1}}$ | $0$ | – |
| $d_B$ | Decay rate coefficient of biomass | $\mathrm{d^{-1}}$ | $4.81 \times 10^{-3}$ | $1.37 \times 10^{-3} - 2.74 \times 10^{-1}$ |
| $d'_B$ | Density-dependent $d_B$ | $\mathrm{g\,mg\,C^{-1}\,d^{-1}}$ | – | $1.37 \times 10^{-2} - 2.74$ |
| $d_E$ | Decay rate coefficient of extracellular enzymes | $\mathrm{d^{-1}}$ | $2.49 \times 10^{-2}$ | $2.74 \times 10^{-3} - 2.74$ |
| $m_E$ | Constitutive enzyme production rate coefficient | $\mathrm{d^{-1}}$ | $1.25 \times 10^{-4}$ | $8.22 \times 10^{-5} - 1.83 \times 10^{-4} \mid 0$ |
| **Partitioning coefficients** | | | | |
| $r_B$ | Recycling efficiency of decayed biomass | 1 | $1.00$ | $0.20 - 1.00$ |
| $y_m$ | Maximal yield | 1 | $0.60$ | $0.01 - 0.80$ |
| $y_B$ | Fraction of uptake going to biomass production | 1 | $y_m$ | $0 < y_B \leq y_m \mid y_m$ |
| $f_I$ | Fraction of input going to SOC | 1 | $0.90$ | $0.50 - 1.00$ |
| $f_D$ | Fraction of decayed biomass going to SOC | 1 | $0.50$ | $0.50 - 1.00$ |
| **Parameter groups** | | | | |
| $\alpha$ | Extracellular enzyme turnover | $\mathrm{d^{-1}}$ | $\alpha = d_E + l_E$ | |
| $\beta$ | Microbial biomass turnover | $\mathrm{d^{-1}}$ | $\beta = d_B + m_E$ | |
| $\eta$ | Parameter group 1 | $\mathrm{d^{-1}}$ | $\eta = (y_m - y_B)d_B + y_m m_E \geq 0$ | |
| $\omega$ | Parameter group 2 | $\mathrm{d^{-2}}$ | $\omega = \alpha\beta - \alpha y_B r_B d_B - \eta d_E \geq 0$ | |

state variables are simultaneously positive and real. Within the physically meaningful equilibrium points, we distinguish three categories:

1. stable, CE2 where all physically meaningful equilibrium points CE3 are stable (i.e., stable node or focus points; Argyris et al., 2015);

2. stable and plausible, where all physically meaningful equilibrium points are stable and also give plausible numerical results;

3. unstable, where all physically meaningful equilibrium points are not stable (i.e., unstable node or focus points; Argyris et al., 2015).

Based on data synthesized by Wang et al. (2013) and on educated guesses, we applied the following conditions for considering results to be plausible: tOC = SOC + DOC + MBC + ENZ $\leq 500\,\mathrm{mgC\,g^{-1}}$ ($= 50\,\%$), DOC/tOC $< 0.01$, MBC/tOC $< 0.05$, and ENZ/MBC $< 0.1$, where tOC indicates the total organic carbon content (the sum of all four carbon pools).

### 2.3.4 Causal loop analysis

In addition to the mathematical analysis of equilibrium points and their stability, we present causal loop diagrams that qualitatively summarize causal links in a system and the feedbacks they create (Haraldsson, 2004). This analysis can help to understand the behavior a system exhibits after a per-

**Table 4.** Summary of analyzed models, respective simplifying assumptions, and types of analysis (ana. – analytical, num. – numerical).

| Model | No. of pools | Kinetics | Simplifying assumptions | | Analysis |
|---|---|---|---|---|---|
| SBE | 3 | $m, f, r, e$ | none | | ana. |
| SDB | 3 | $m \times m, f \times f, r \times f$ | $L_S = L_E = 0$; | $y_B = y_m$ | ana. |
| SDBE | 4 | $m \times m, f \times f, r \times f$ | $L_S = L_E = 0$ CE4 $\left\{ \begin{array}{l} y_B = y_m \\ m_E = 0 \end{array} \right.$ | | ana. + num. / num. |

turbation around an equilibrium point. In a causal loop diagram, causal connections are depicted by arrows, tying a cause (at the tail of the arrow) to its direct effect (at the head of the arrow) (Haraldsson, 2004). The sign of the causal relation (+ or −) depends on whether an isolated change in one element causes another to change in the same (+) or opposite (−) direction in relation to the initial change (relative to the unchanged state) (Haraldsson, 2004; Richardson, 1986). For example, a decrease in the microbial uptake rate would lead to relatively less microbial biomass (compared to the case where the uptake rate had not changed), describing a positive causal relation. Closed loops with zero or an even number of negative interactions are denoted as positive or reinforcing feedback loops $R$, and closed loops with a odd number of negative interactions are denoted as negative or balancing feedback loops $B$ (Haraldsson, 2004; Richardson, 1986).

## 3 Results

We analyzed three model versions with different model structures (number of state variables and/or non-linearities; Table 4, Sect. 2.2). We first present analytical results for the simpler three-pool models with one (SBE) and two nonlinear terms (SDB), followed by analytical and numerical results for the four-pool SDBE model (Table 4). We use causal loop diagrams to qualitatively interpret these results.

### 3.1 SBE model: neglecting DOC dynamics

#### 3.1.1 Steady-state solutions

For all kinetic descriptions of the depolymerization rate (Table 2), the three-pool SBE model has two equilibrium points (EPs). Of these, one is an "abiotic" equilibrium $Q_0$, where only SOC exists and where microbial biomass and extracellular enzymes are zero; i.e., $Q_0 = (S_0^*, 0, 0)$. Here, the asterisk indicates a state variable at steady state, and the subscript 0 signifies the abiotic solution. In turn, for each kinetic, there exists an alternative "biotic" equilibrium point with non-zero microbial biomass and extracellular enzymes; i.e., $Q_{1,i} = (S_{1,i}^*, B_{1,i}^*, E_{1,i}^*)$. The steady-state solutions for these equilibria depend on the $i$th formulation used to describe $P_i$. All solutions are reported in Table 5, where, for convenience, parameters have been combined into parameter groups: extracellular enzyme turnover $\alpha > 0$, microbial biomass turnover $\beta > 0$, and parameter groups $\eta \geq 0$ and $\omega \geq 0$ (Table 3).

While the abiotic equilibrium point is always positive, the biotic one can only be positive (and thus physically meaningful) if

$$S_{1,i}^* < S_0^* \rightarrow l_S S_{1,i}^* < I, \tag{20}$$

i.e., if the linear SOC loss rate is smaller than SOC inputs. Note that, for $f$ and $e$ kinetics, additional conditions apply for positivity.

If the abiotic loss of SOC is neglected (i.e., $l_S = 0$), the abiotic equilibrium point does not exist (SOC would accumulate at the constant rate $I$), and, in accordance with Eq. (20), the biotic equilibrium is always physically meaningful.

#### 3.1.2 Stability analysis

To analyze whether a physically meaningful equilibrium point is also stable, we apply the Routh–Hurwitz criterion to the Jacobian matrix $J_i^{\mathrm{SBE}}$ (Eq. A1 in the Appendix) evaluated at the $k$th equilibrium point ($J_i^{\mathrm{SBE}}|_{*,k}$) – either the abiotic ($k = 0$) or the biotic equilibrium ($k = 1$) (detailed in Sect. A1 in the Appendix and in Sect. S2.1 in the Supplement). From this, we find that the stability of a physically meaningful abiotic equilibrium is conditional on

$$\left. \frac{\partial P_i}{\partial E} \right|_{*,0} < \frac{\alpha \beta}{\eta}, \tag{21}$$

whereas all physically meaningful biotic-equilibrium points of the three-pool SBE model are also stable.

Figure 2a shows a simplified causal loop diagram of the SBE model (sparing all loss and decay terms) that can help to understand the dynamic behavior of the model after a perturbation around an equilibrium point. The reinforcing loop $R_1$ describes the increase in microbial biomass with an increasing depolymerization rate ($\propto$ uptake rate), leading to an increased ENZ production rate; an increased ENZ concentration; and, consequently, a further increasing depolymerization rate. This reinforcing effect is dampened by the balancing loops $B_1$ (the depletion of SOC by depolymerization) and $B_2$ (the carbon cost of ENZ production). The reinforcing loop $R_2$ exists only if inducible ENZ production is considered ($y_B < y_m$; higher depolymerization stipulates the production of more extracellular enzymes, which promote depolymerization). $R_1$ and $R_2$ are not independent of each other

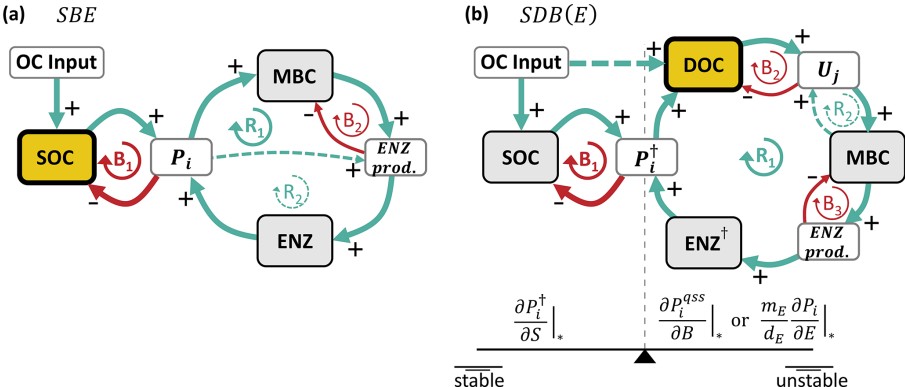

**Figure 2.** Simplified causal loop diagrams of the SBE (**a**) and SDB and SDBE models (**b**). Green arrows marked with "+" indicate positive interactions, and red arrows marked with "−" indicate negative interactions. $R$ signifies reinforcing loops, and $B$ signifies balancing loops. Pools are marked by gray boxes, and rates are marked by white boxes. The microbial growth substrate is highlighted in yellow with a thick outline. In (**a**), the dashed green line (at $R_2$) indicates the effect of inducible ENZ production and vanishes if only constitutive ENZ production is considered. In (**b**), the dashed green line (at $R_2$) indicates the effect of biomass-dependent non-linear uptake kinetics and vanishes for biomass-independent uptake kinetics. In the SDB (SDBE) model $P_i^\dagger$ and ENZ$^\dagger$ signify $P_i^{qss}$ ($P_i$) and $E_{i \times j}^{qss}$ ($E_{i \times j}^*$). The seesaw in (**b**) illustrates the balance between the partial derivatives in Eqs. (25) and (26) and how they affect stability. See Sect. 2.3.4 for details on how to read the causal loop diagrams.

and have to obey mass balance – i.e., per unit of uptake, an increase in inducible ENZ production ($\propto y_m - y_B$) can only be achieved by reducing the build-up of microbial biomass (lowering $y_B$). An extreme case of this is $y_B = 0$; in this case, no microbial biomass is produced, and only $B_1$ and $R_2$ remain. From the stability analysis (Sect. A1 and Sect. S2.1 in the Supplement), we obtained no conditionality on the stability of physically meaningful equilibrium points in the SBE model. Thus, for any proportion of constitutive vs. inducible ENZ production, all physically meaningful biotic equilibria are also stable. That is, the dynamic behavior of the model after a perturbation around its equilibrium point is dominated by the balancing feedbacks, ensuring a convergence back to the equilibrium point.

### 3.1.3 Exclusive stability of either abiotic or biotic equilibrium

We recall that, for the biotic equilibrium to be physically meaningful, it is required that $S_{1,i}^* < S_0^*$ (Eq. 20); whereas, for the abiotic equilibrium point to be stable, it is required by Eq. (21) that $\frac{\partial P_i}{\partial E}\big|_{*,0} < \frac{\alpha \beta}{\eta}$. This condition translates for, e.g., multiplicative kinetics to

$$\frac{\partial P_m}{\partial E}\bigg|_{*,0} = v_m^p S_0^* < \frac{\alpha \beta}{\eta} \rightarrow S_0^* < \frac{\alpha \beta}{v_m^p \eta} = S_{1,m}^*. \quad (22)$$

This means that, when the biotic equilibrium is physically meaningful, the abiotic equilibrium is unstable and vice versa. Therefore, no region in the parameter space yields a physically meaningful bi-stability in which biotic and abiotic equilibria are simultaneously physically meaningful and stable. This holds for all evaluated kinetics (see Sect. S2.1 in the Supplement for the remaining analytical derivations).

### 3.2 SDB model: neglecting ENZ dynamics

#### 3.2.1 Steady-state solutions

Table 6 reports the steady-state solutions of the three-pool SDB model, where, for convenience, parameters were further grouped into

$$\gamma_{i \times j} = f_D r_B d_B y_m + \pi f_I \frac{I}{I - l_D D_{1,i \times j}^*} \quad (23)$$

and

$$\pi = \frac{1}{d_E} \omega(l_E = 0, y_B = y_m) = \underbrace{(1 - y_m)m_E}_{>0}$$
$$+ \underbrace{(1 - y_m r_B)d_B}_{>0} > 0. \quad (24)$$

Because an abiotic equilibrium point exists only for $l_S > 0$ (Table 6), we only evaluate the stability of the biotic equilibrium and drop "1" from the subscript for conciseness. Biotic steady states can only be physically meaningful for $I > l_D D_{i \times j}^*$, meaning that the DOC leaching flux cannot be larger than the total OC input flux. With $f \times f$ and $r \times f$ kinetics, it is additionally required that $\beta < y_m v_f^u$, implying that the maximal per-biomass assimilation rate $y_m v_f^u$ must be larger than the microbial biomass turnover ($\beta = m_E + d_B$; Table 3). For $f \times f$ kinetics, it is additionally required that $\frac{d_E}{y_m v_f^p m_E} \gamma_{f \times f} < 1$ for steady states to be positive.

In the absence of DOC leaching (for $l_D = 0$), with $m \times m$ and $f \times f$ kinetics, $S_{i \times j}^*$ becomes independent of $I$ (by Eq. 23), while $B_{i \times j}^*$ and $E_{i \times j}^*$ are linear functions of

**Table 5.** Summary of steady-state solutions of the three-pool SBE model for different kinetics of depolymerization. The "biotic" equilibrium solutions for microbial biomass and extracellular enzymes have the same form for any chosen kinetic.

| | Kinetic ($i$) | $S_{k,i}^*$ | $B_{k,i}^*$ | $E_{k,i}^*$ |
|---|---|---|---|---|
| Abiotic ($k=0$) | $i$ | $\frac{I}{l_S}$ | 0 | 0 |
| Biotic ($k=1$) | $m$ | $\frac{\alpha\beta}{v_m^p\eta}$ | | |
| | $f$ | $\frac{\alpha\beta}{v_f^p\eta}\frac{K_f^p}{1-\frac{\alpha\beta}{v_f^p\eta}}$ | $\frac{\alpha y_B}{\omega}l_S\left(S_0^*-S_{1,i}^*\right)$ | $\frac{\eta}{\alpha y_B}B_{1,i}^*$ |
| | $r$ | $\frac{\alpha\beta}{v_r^p\eta}\frac{K_r^p\omega+I\eta}{\omega+l_S\frac{\alpha\beta}{v_r^p}}$ | | |
| | $e$ | $\frac{\alpha\beta}{v_e^p\eta}\frac{K_e^p\omega+I\eta}{\omega-\omega\frac{\alpha\beta}{v_e^p\eta}+l_S\frac{\alpha\beta}{v_e^p}}$ | | |

**Table 6.** Summary of steady states for the three-pool SDB and four-pool SDBE model for different kinetics of depolymerization and uptake. The abiotic steady state is only defined for $l_S > 0$. Biotic steady states are given for $l_E = l_S = 0$ and $y_B = y_m$. $E_{k,i\times j}^\dagger$ signifies $E_{k,i\times j}^{\mathrm{qss}}$ or $E_{k,i\times j}^*$ in the SDB and SDBE models, respectively. The biotic-equilibrium solutions for microbial biomass and enzymes have the same form for all chosen kinetics.

| | Kinetics | $S_{k,i\times j}^*$ | $D_{k,i\times j}^*$ | $B_{k,i\times j}^*$ | $E_{k,i\times j}^\dagger$ |
|---|---|---|---|---|---|
| Abiotic ($k=0$) | $i\times j$ | $f_I\frac{I}{l_S}$ | $(1-f_I)\frac{I}{l_D}$ | 0 | 0 |
| Biotic ($k=1$) | $m\times m$ | $\frac{d_E}{y_m v_m^p m_E}\gamma_{m\times m}$ | $\frac{\beta}{y_m v_m^u}$ | | |
| | $f\times f$ | $\frac{d_E}{y_m v_f^p m_E}\gamma_{f\times f}\frac{K_f^p}{1-\frac{d_E}{y_m v_f^p m_E}\gamma_{f\times f}}$ | $K_f^u\frac{\beta}{y_m v_f^u-\beta}$ | $\frac{y_m}{\pi}\left(I-l_D D_{1,i\times j}^*\right)$ | $\frac{m_E}{d_E}B_{1,i\times j}^*$ |
| | $r\times f$ | $\frac{d_E}{y_m v_r^p m_E}\gamma_{r\times f}\left(K_r^p+E_{1,r\times f}^\dagger\right)$ | $K_f^u\frac{\beta}{y_m v_f^u-\beta}$ | | |

$I$. In contrast, for $l_D = 0$, $S_{r\times f}^*$ is linearly dependent on OC input $I$. For $l_D > 0$, $S_{m\times m}^*$ is a function of $\frac{I}{I-l_D D_{m\times m}^*}$ (Eq. 23), causing $S_{m\times m}^*$ to decline with increasing inputs as $\frac{I}{I-l_D D_{m\times m}^*} \to 1$ for $I \gg l_D D_{m\times m}^*$. Only in $S_{r\times f}^*$ does $I$ still appear in a linear term for $l_D > 0$ as well.

## 3.2.2 Stability analysis

The Jacobian matrix of the SDB model around the biotic equilibrium, $J_{i\times j}^{\mathrm{SDB}}\big|_*$, is given by Eq. (A10). Evaluating the respective coefficients of the characteristic polynomial and the requirement that $a_1 a_2 - a_3 > 0$ yields a cumbersome sufficient and necessary condition for the stability of the biotic EPs of the SDB model (Eqs. A11–A13; see details in Sect. A2 and Sect. S2.2 in the Supplement). The appearance of such a conditional statement means that, in contrast to the SBE model, the SDB model can have physically meaningful but unstable EPs (i.e., if the conditions described by Eqs. A11–A13 do not hold). A perturbation around such an unstable EP will cause the system to diverge from the EP. In this case, the biotic pools (MBC and quasi-steady-state ENZ) will collapse, while DOC will reach a steady state as $D_{0,i\times j}^*$ (for $l_D > 0$), and SOC will accumulate indefinitely.

The obtained sufficient and necessary condition given by Eqs. (A11)–(A13) does not allow for easy interpretation or application. We thus propose a sufficient (i.e., a more conservative or strict) condition for stability that is easier to trace analytically as follows:

$$\frac{\partial P_i^{\mathrm{qss}}}{\partial S}\bigg|_* + y_m f_D r_B d_B \geq y_m \frac{\partial P_i^{\mathrm{qss}}}{\partial B}\bigg|_*. \tag{25}$$

This sufficient condition for the stability of the biotic-equilibrium points of the SDB model holds for all cases relevant to our analysis (Sect. S2.2 in the Supplement). Described in words, this condition requires the depolymerization rate to be less sensitive to a change in microbial biomass than to a proportional change in SOC. Figure 2b illustrates this relation in a simplified causal loop diagram. The reinforcing loop $R_1$ causes the depolymerization rate ($P_i^{\mathrm{qss}}$) to increase as the (quasi-steady-state) ENZ concentration increases (quantified by $\frac{\partial P_i^{\mathrm{qss}}}{\partial B}\big|_*$). This then increases the DOC concentration and uptake and ultimately causes a further increase in microbial biomass and (quasi-steady-state) ENZs. This positive feedback is accelerated by an additional reinforcing feedback loop ($R_2$) that causes uptake to further increase as microbial biomass increases. The balancing loops

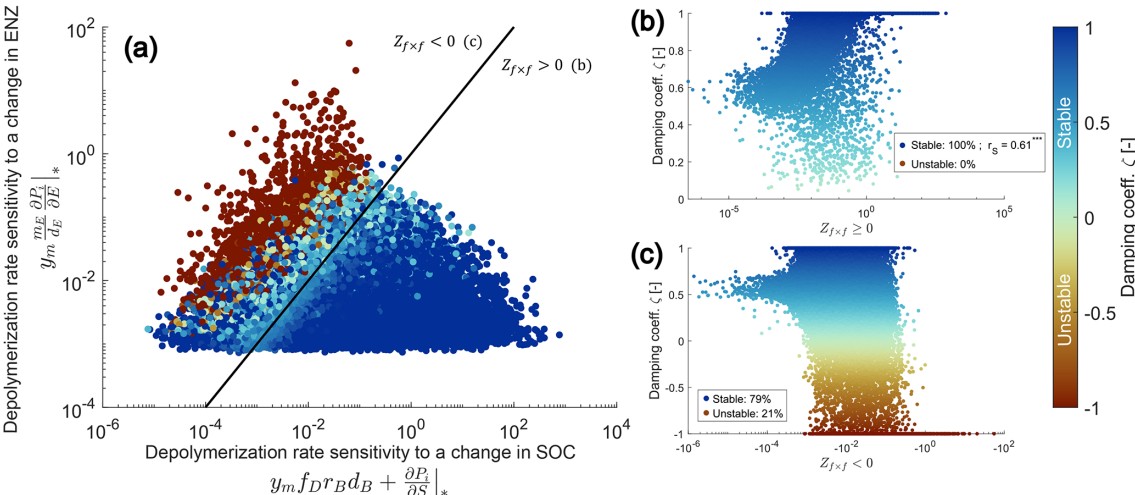

**Figure 3.** Numerical evaluation of the proposed sufficient condition for stability of the SDBE model with $f \times f$ kinetics and constitutive ENZ production. The proposed stability condition $Z_{f \times f} \geq 0$ (Eq. 26) is derived from an analytical evaluation of the simpler SDB model. A total of 100 000 Monte Carlo simulations within the parameter ranges given in Table 3 were produced. Panel **(a)** illustrates the separation of all physically meaningful equilibrium points by the positive and negative terms of $Z_{f \times f}$. Points on and below the black $1 : 1$ line (indicating $Z_{f \times f} = 0$) fulfill the condition $Z_{f \times f} \geq 0$. The color code indicates the value of the damping coefficient $\zeta$. Panels **(b)** and **(c)** show values of $\zeta$ vs. values of $Z_{f \times f}$ (for $Z_{f \times f} \geq 0$ in **b** and for $Z_{f \times f} < 0$ in **c**). Legends in **(b)** and **(c)** state the percentages of stable and unstable equilibrium points; $r_S$ in **(b)** gives the Spearman rank correlation coefficient (*** indicates significant at $p < 0.005$).

$B_2$ and $B_3$, respectively, describe the depletion of DOC with increasing uptake and the reduction of biomass as more extracellular enzymes are produced. Lastly, the balancing loop $B_1$ causes SOC to change in the opposite direction compared to the depolymerization rate (i.e., SOC is depleted as depolymerization increases, and relatively more SOC remains as depolymerization decreases), counteracting the initial change in $P_i^{\mathrm{qss}}$ (quantified by $\frac{\partial P_i^{\mathrm{qss}}}{\partial S}\big|_*$). Therefore, the sufficient stability condition in Eq. (25) can be interpreted in the sense that the negative feedback $B_1$ must be quantitatively stronger than the positive feedback $R_1$ (by some factor $y_m$ and buffered by a constant term; Eq. 25).

In essence, the positive feedback $R_1$ can drive the system to overshoot or collapse: e.g., if the microbial biomass or (quasi-steady-state) ENZ concentration happens to decrease due to a perturbation, this will reduce the depolymerization rate and, following the positive feedback, will result in further reduced MBC and (quasi-steady-state) ENZ concentrations. The biotic pools would collapse, and the system would not be able to recover to its initial equilibrium (i.e., be unstable). Only if the entailing accumulation of SOC increases the depolymerization rate more than it is reduced by the depletion of (quasi-steady-state) ENZs will the system be able to recover and retain the biotic components (i.e., be stable).

We note that, for linear uptake kinetics (i.e., $U_1 = v_l^u D$), additional terms in the necessary and sufficient conditions can help to fulfill the Routh–Hurwitz criterion for stability (see details in Sect. A2). In the causal loop diagram, linear uptake kinetics remove the positive feedback between uptake

rate and biomass ($R_2$ in Fig. 2b vanishes). Although we could not show this analytically, numerical evaluation showed that, for the chosen parameter spaces (Table 3) with linear uptake kinetics $U_1$, the physically meaningful EPs of the SDB model were always stable (Fig. S3c–d in the Supplement).

### 3.3 SDBE model: full archetypal model

The four-pool SDBE model with $L_S = L_E = 0$ and $y_B = y_m$ has the same steady-state solutions as the three-pool SDB model, but now the solution for the ENZ pool is denoted as $E_{i \times j}^*$ because ENZ is not considered to be at a quasi-steady state as in the SDB model (Table 6).

#### 3.3.1 Analytical stability analysis

In the four-pool SDBE model, the coefficients of the characteristic polynomial of $J_{i \times j}^{\mathrm{SDBE}}\big|_*$ (Eq. A17) remain analytically tractable (Sect. S2.3 in the Supplement). The trace of $J_{i \times j}^{\mathrm{SDBE}}\big|_*$ is always negative (and, thus, $-\mathrm{tr}(J_{i \times j}^{\mathrm{SDBE}}\big|_*) = a_1 > 0$), and its determinant is always positive ($\det(J_{i \times j}^{\mathrm{SDBE}}\big|_*) = a_4 > 0$). However, the additional Routh–Hurwitz criterion for the $4 \times 4$ matrix $J_{i \times j}^{\mathrm{SDBE}}\big|_*$ (given by $a_1 a_2 a_3 - a_3^2 - a_1^2 a_4 > 0$) becomes analytically intractable. Despite this additional complexity, we can still draw some conclusions based on similarities between the SDB and SDBE models. Considering that, in the SDBE model, ENZ dynamics are explicitly represented (and, thus, e.g., $\frac{\partial P_i^{\mathrm{qss}}}{\partial B} \to \frac{m_E}{d_E} \frac{\partial P_i}{\partial E}$), similar conditions emerge for the positivity of the coefficients of the characteristic polynomial, as in the SDB model (Sect. S2.3 in the

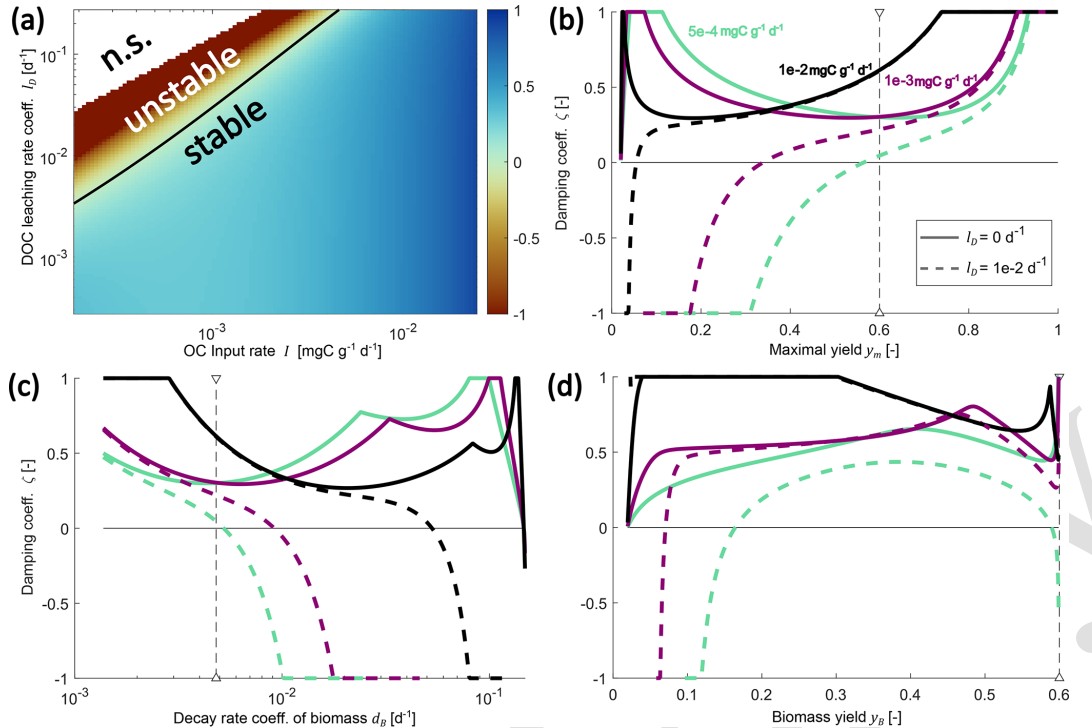

**Figure 4.** Changes in the damping coefficient with changes in environmental controls **(a)** and some microbial physiology parameters **(b–d)** in the SDBE model with $f \times f$ kinetics. Panel **(a)** shows the damping coefficient as a function of $I$ and $l_D$, with all other parameters held at their baseline values (Table 3). The black line indicates $\zeta = 0$ (n.s. indicates no physically meaningful solution). Panels **(b)–(d)** show variation in the damping coefficient for different combinations of $I$ and $l_D$ values. Different line styles indicate scenarios with different DOC leaching rate coefficients: solid lines indicate $l_D = 0$, and dashed lines indicate $l_D = 1 \times 10^{-2}\,\mathrm{d}^{-1}$. Different line colors indicate scenarios with different OC input rates: turquoise lines indicate $I = 5 \times 10^{-4}\,\mathrm{mg\,C\,g}^{-1}\,\mathrm{d}^{-1}$, violet lines indicate $I = 1 \times 10^{-3}\,\mathrm{mg\,C\,g}^{-1}\,\mathrm{d}^{-1}$, and black lines indicate $I = 1 \times 10^{-2}\,\mathrm{mg\,C\,g}^{-1}\,\mathrm{d}^{-1}$. In **(b)** and **(c)**, the baseline model with constitutive production of extracellular enzymes is used, and $y_m$ and $d_B$ are varied, respectively. Baseline parameter values are indicated by vertical dashed lines. In **(d)**, instead, only inducible production of extracellular enzymes is considered, and $y_B$ is varied. Note the logarithmic scaling of $x$ and $y$ axes in panel **(a)** and of the $x$ axis in panel **(c)**.

Supplement). Based on these similarities, we propose that the sufficient condition for stability found for the three-pool SDB model might also hold in the SDBE model. This proposed sufficient condition is given by

$$Z_{i \times j} = \left.\frac{\partial P_i}{\partial S}\right|_* + y_m f_D r_B d_B - y_m \frac{m_E}{d_E}\left.\frac{\partial P_i}{\partial E}\right|_* ; \quad Z_{i \times j} \geq 0. \quad (26)$$

The simplified causal loop diagram of the SDBE model in Fig. 2b gives rise to the same interpretation of this condition as in the SDB model. In the following, we confirm that this condition holds in the SDBE model via numerical analysis.

### 3.3.2 Numerical stability analysis

**Testing the sufficient condition for stability**

We produced 100 000 Monte Carlo simulations and computed the damping coefficient $\zeta$ (Eq. 19) to numerically evaluate the stability of equilibrium points in the SDBE model. Within the sampled parameter space (Table 3), physically

meaningful equilibrium points for which $Z_{i \times j} \geq 0$ (Eq. 26) also always had $\zeta > 0$ and were thus stable (Fig. 3a–b, Fig. S1 in the Supplement). Damping coefficients with $\zeta \leq 0$ were only observed when $Z_{i \times j} < 0$ (that is for points above the black line in Fig. 3a or red points in Fig. 3c). While a total of 46 932 evaluated equilibrium points were physically meaningful and stable, fewer than half of these (22 386) also fulfilled the condition $Z_{i \times j} \geq 0$. In turn, the majority of these equilibrium points (24 546) were stable despite contradicting this condition – i.e., the condition given by $Z_{i \times j} \geq 0$ is very conservative. In case the condition is fulfilled, the value of $Z_{i \times j}$ correlates well with the value of $\zeta$, meaning that, for larger $Z_{i \times j}$, oscillations are generally more dampened (Fig. 3b).

**Changing environmental conditions, microbial physiology, and stability**

Even for the simplified sufficient condition $Z_{i \times j} \geq 0$, analysis is cumbersome for kinetics other than $m \times m$ (Sect. S2.3 in

**Table 7.** Indication of thresholds in parameter values for stability of the SDBE model. Analysis is based on exclusively varying one parameter while keeping all others at their baseline value (Table 3, Fig. 4, Fig. S2 in the Supplement). The meaning of the different thresholds is illustrated in the schematics on the right. This analysis applies to all evaluated combinations of kinetics ($m \times m$, $f \times f$, $r \times f$) using constitutive ENZ production, except for the analysis of $y_B$, where we considered only inducible ENZ production. Note: ($i = m, f, r$), ($j = m, f$).

| Process | Parameter | Threshold Lower | Threshold Upper | Threshold None |
|---|---|:---:|:---:|:---:|
| Depolymerization | $v_i^p$ | X | | |
| | $K_i^p$ | | X | |
| Assimilation | $v_j^u$ | X | | |
| | $K_f^u$ | | X | |
| | $y_m$ | X | | |
| ENZ production | $m_E$ | X | | |
| | $y_B$ | X | X | |
| Decay | $d_B$ | | X | |
| | $d_E$ | | X | |
| Mass balance losses | $r_B$ | X | | |
| | $l_D$ | | X | |
| OC Input | $I$ | X | | |
| SOC–DOC partition | $f_I$ | | X | |
| | $f_D$ | | | X |

the Supplement). We thus varied specific parameters individually and evaluated their effect on the numerically computed damping coefficient $\zeta$.

Keeping all microbial and enzymatic parameters constant (set to their baseline value – that is for constitutive ENZ production; Table 3), stability depends on the environmental control parameters $l_D$ and $I$ (Fig. 4a). At baseline parameter values, most environmental conditions yield stable EPs, but strong oscillations occur around these EPs (damping coefficient $\zeta < 1$). As conditions become less favorable and as either $l_D$ increases and/ or $I$ decreases, equilibrium points can become unstable ($\zeta < 0$).

Next, we analyzed the influence of individual microbial parameter values on the stability of EPs for a number of scenarios defined by combinations of $I$ and $l_D$ (Fig. 4b–d; Table 7). Generally, if DOC leaching is neglected (solid lines, Fig. 4b–d), the variation in just one parameter rarely leads to unstable equilibria (only at very high microbial decay rates $d_B$). In contrast, if DOC leaching occurs, variation in key physiological parameters can lead to a transition from stable to unstable EPs. This happens, e.g., as $y_m$ becomes too low or $d_B$ becomes too high – i.e., for specific environmental conditions, there are lower threshold values for $y_m$ and upper thresholds values for $d_B$ beyond which equilibria become unstable (Fig. 4b–c, Table 7). With the exception of $f_D$, the partitioning of decayed microbial biomass between SOC and DOC, all parameters show such a threshold within the explored ranges (Fig. S2 and Sect. S2.3 in the Supplement).

Using the alternative description of inducible production of extracellular enzymes, the stability behavior with respect to changes in $y_B$ is more varied (Fig. 4d). As for $y_m$, there are lower $y_B$ threshold values below which steady states become unstable. However, there can also be upper thresholds for $y_B$ above which too few enzymes are being produced to ensure sufficient C acquisition.

In summary, by varying only individual parameters, instabilities can arise when assimilation, depolymerization, or ENZ production are too low or when abiotic C losses are too high (Fig. 4b–d, Fig. S2 in the Supplement, Table 7). These results are in line with the analytic analysis of the sufficient condition (Eq. 26) for $m \times m$ kinetics (Sect. S2.3 in the Supplement).

**Density-dependent mortality**

Georgiou et al. (2017) proposed a density-dependent formulation of the microbial decay rate ($D'_B = d'_B B^b$, with $1 < b \leq 2$) as an alternative to the conventional linear decay term that yields mostly stable non-oscillatory behavior (note that this formulation causes both microbial mortality and maintenance respiration to be density-dependent). For our SDBE model with DOC leaching, we could only find an analytical steady-state solution for $m \times m$ kinetics and $b = 2$. In this case, the density-dependent formulation could vastly alleviate the previously observed instability and could result in damping coefficients for plausible equilibrium points close to 1 for most of the explored parameter spaces (Fig. S4a in

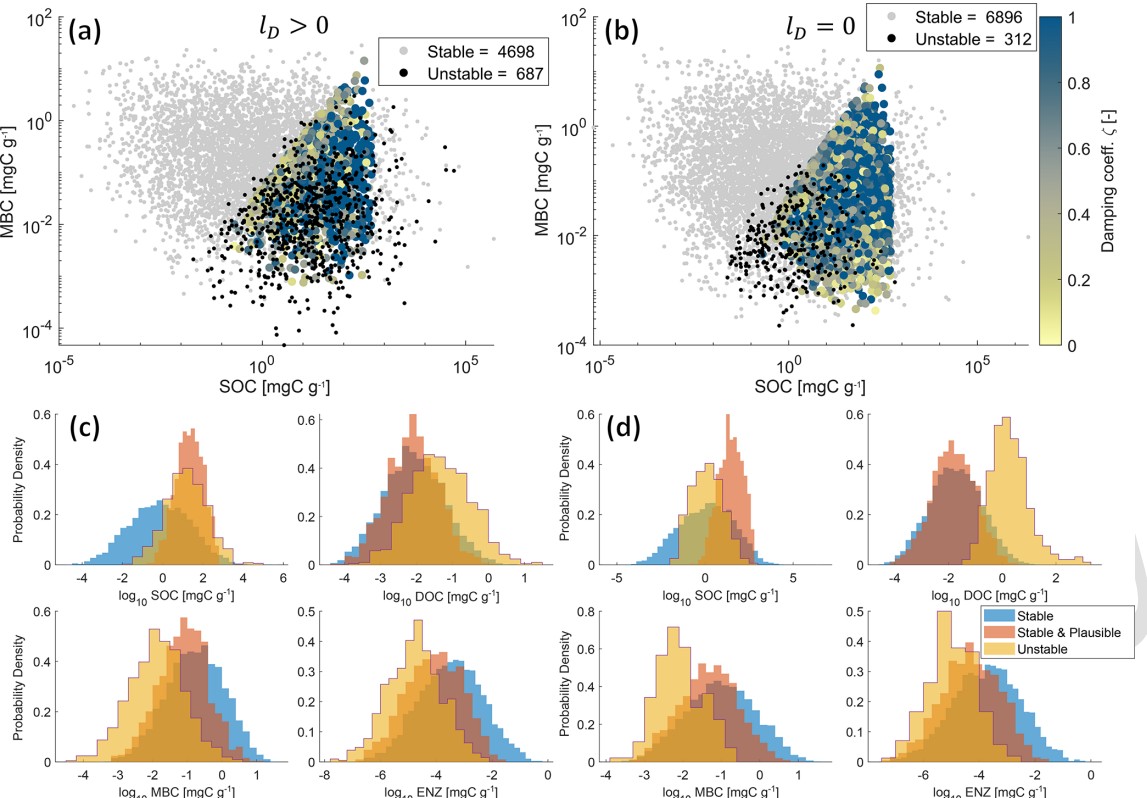

**Figure 5.** Physically meaningful (positive and real) steady-state solutions from 10 000 Monte Carlo simulations of the SDBE model with $f \times f$ kinetics and constitutive ENZ production. Scatterplots of MBC vs. SOC concentrations are shown for $l_D > 0$ **(a)** and $l_D = 0$ **(b)**. Color-coded points are stable and plausible steady-state solutions, with the color code indicating the value of the damping coefficient. Gray points are stable but not plausible steady-state solutions, and black points are physically meaningful but unstable steady-state solutions. Legends indicate the numbers of physically meaningful and stable (stable + stable and plausible) or unstable EPs. Plot groups **(c)** and **(d)** show empirical probability density functions of each state variable for stable, stable and plausible, and unstable physically meaningful steady-state solutions for $l_D > 0$ **(c)** and $l_D = 0$ **(d)**, respectively. Note that SOC contents $> 1000 \, \mathrm{mg \, C \, g^{-1}}$ are mathematically possible but unphysical model outcomes as we neglect soil volume changes.

the Supplement). However, some physically meaningful but unstable EPs were still observed. Only with negligible DOC leaching did physically meaningful but unstable equilibrium points vanish completely. This was numerically tested for $m \times m$, $f \times f$, and $r \times f$ kinetics (Fig. S4b–d in the Supplement).

**Instability and predicted organic carbon pools**

Figure 5a–b illustrate the joint distributions of physically meaningful SOC and MBC pools in the $f \times f$ model for scenarios where DOC leaching is either considered (Fig. 5a) or neglected (Fig. 5b) (for 10 000 Monte Carlo simulations within the parameter ranges given in Table 3). By accounting for DOC leaching, just about half of the simulations yield physically meaningful EPs – of which most (4698) were also stable, but more than 10 % (687) were unstable. Neglecting DOC leaching increases the total number of physically meaningful results (6896) and simultaneously reduces the relative share of physically meaningful but unstable results to

$< 5\%$ (312). In both scenarios, most simulations yield implausible results for steady-state C stocks – e.g., MBC being larger than SOC. However, in both scenarios, unstable EPs largely overlap with stable and plausible outcomes in the SOC–MBC solution space. This is also evident in the empirical probability density functions of all four state variables (Fig. 5c–d). Especially if DOC leaching is considered ($l_D > 0$, Fig. 5c), values of plausible and unstable steady-state SOC pools largely overlap. In contrast, the distributions of plausible and unstable steady-state pool sizes of MBC, DOC, and ENZ do not overlap as closely as those for SOC. These distinctions are amplified in the cases where DOC leaching is neglected ($l_D = 0$, Fig. 5d), in particular for DOC.

## 4 Discussion

### 4.1 Model structure matters: standard microbial-explicit SOC models can have unstable equilibria

Microbial-implicit models with linear decomposition kinetics are stable as long as there is no inert pool, mass conservation applies, and rates are proportional to the amount of carbon in the donor pool (Sierra and Müller, 2015). It is more difficult to define general stability criteria for nonlinear models, whose structure and type of non-linearity affects the model behavior around the equilibrium state. Manzoni and Porporato (2007) and Raupach (2007) showed analytically that the non-trivial steady states of two-pool models, consisting of a substrate pool and a microbial pool (denoted as a "harvester" system in Raupach, 2007), are always stable for multiplicative and forward Michaelis–Menten kinetics (but only under the assumption that the input to the substrate ($I$) is a constant; Raupach, 2007). We show here that the same is also true for all physically meaningful, nontrivial (biotic) equilibrium points if a third pool representing extracellular enzymes is added (SBE model). This result holds irrespective of the kinetic laws used to describe SOC depolymerization and whether ENZ production is considered to be constitutive, inducible, or a combination of both.

Interestingly, by introducing a second non-linear term, Raupach (2007) found that unstable equilibria could emerge in their two-pool model. In contrast, Wang et al. (2014, 2016) demonstrated for several versions of a three-pool (litter–SOC–microbes) model with two non-linearities (microbial degradation and subsequent uptake of litter and SOC) that the equilibrium points of these models were always stable. An underlying assumption in these models was that the available substrate pool (similarly to what we described as DOC) was at a quasi-steady state. Our derivation of the SBE model follows a similar simplification – and also does not yield unstable behavior. By contrast, unstable equilibrium points are possible in our three- and four-pool model versions with two non-linearities that explicitly consider DOC (SDB and SDBE models; yellow boxes in Fig. 2). Whether equilibrium points in microbial-explicit SOC models can become unstable is thus not dependent on the number of pools or the number of non-linearities per se but is rather dependent on the combination of non-linearities, the coupling of different pools and rates, and what feedbacks they create.

Comparing our three- and four-pool models to the simpler two-pool model analyzed by Raupach (2007) can help to understand why instability can occur in these models. Briefly, their model describes human consumption of a food resource but is similar in structure to our models (analogous terms in our models are given in brackets): the resource (SOC in SBE and DOC in SDB and SDBE models) is taken up by the human consumer (microbes) and thereby depleted. The uptake process is always described as a non-linear term, equivalent

to our description of $U_j$. Raupach (2007) analyzed two different cases with respect to the resupply of the resource ($I$ in SBE and $(1 - f_I)I + P_i^\dagger$ in SDB and SDBE): (1) resupply is independent of the available resource, and (2) resupply is dependent on the resource itself. The first case is similar to our SBE model, where the resource SOC is replenished by the external input $I$ and is thus independent of the SOC availability itself. In these cases, the biotic (resource–human coexistence in Raupach, 2007) equilibrium is always stable if it is physically meaningful. In turn, the second case can be compared to our SDB and SDBE models: unless the external input to DOC is very high (for low $f_I$), the replenishment of the resource DOC is dominated by the depolymerization rate $P_i^\dagger$ – which, via the positive feedback loop $R_1$, is dependent on the available DOC (compare Fig. 2b). These models can have unstable biotic equilibria (Raupach, 2007). Therefore, the (more or less direct) dependency of the resource resupply on the abundance of the resource itself can be identified as the root cause that allows for instability in these models.

We further tested this hypothesis by setting $f_I = 0$ in the SDBE model (i.e., all external input goes into DOC directly), bypassing the dependency of DOC replenishment on depolymerization. In line with our expectation (and the analytical solution of $Z_{m \times m}$ and $Z_{f \times f}$ for $f_I = 0$ in Sect. S2.3 in the Supplement), this effectively prevented the occurrence of unstable EPs (Fig. S5 in the Supplement).

### 4.2 Avoiding instability

Our analysis of different model structures and their stability behavior points to three approaches to avoid unstable EPs in microbial-explicit SOC models:

1. *Model structure.* Avoid positive feedback coupling between the microbial growth substrate (here DOC) available for uptake and its resupply.

2. *Kinetic formulations.* Avoid accelerated depletion of DOC by reducing the dependency of uptake on microbial biomass.

3. *Parameter values.* Choose parameter values so that the sufficient and/or necessary conditions for stability are met.

The first approach is commonly taken in models that assume DOC to be in a quasi-steady state (e.g., Wang et al., 2014, 2016) but might have shortcomings in cases where DOC dynamics become important – e.g., if drying–rewetting dynamics or leaching are relevant (also see Wutzler and Reichstein, 2013 for a discussion on timescale dependence of appropriate abstraction level). If DOC leaching is not considered to be a relevant process, neglecting this process but keeping a dynamic description of DOC can already considerably reduce the likelihood of unstable EPs.

The second approach is used, e.g., in models that assume DOC uptake to be independent of microbial biomass but de-

pendent on the availability of DOC (e.g., Manzoni et al., 2014). Alternatively, using, for instance, reverse Michaelis–Menten kinetics to describe microbial uptake can dampen oscillations (Wang et al., 2016). Since uptake kinetics using reverse Michaelis–Menten or the ECA formulation become similar to linear uptake kinetics at relatively high concentrations of microbial biomass, they could also help to alleviate instability issues under these conditions. However, when using forward Michaelis–Menten kinetics for decomposition and DOC assimilation, the half-saturation constants can attain large values after data assimilation, suggesting that decomposition kinetics might be approximated by multiplicative rather than linear kinetics, at least in large-scale model applications (Tao et al., 2024b).

Lastly, the third approach might seem straightforward as we could expect parameter values calibrated with measurement data to yield both stable and plausible EPs. However, our numerical simulations indicated that, especially if DOC leaching is considered, calibrating parameter values with SOC and microbial biomass data alone could still lead to plausible yet unstable EPs (Fig. 5). While data on carbon contents in the extracellular enzyme pool are still not available, combining microbial biomass data with quantitative data on DOC pools (as in, e.g., Wang et al., 2013) could help to avoid calibration according to parameter values that lead to unstable EPs. Moreover, stability criteria can be obeyed in various other ways, e.g., by considering correlations between parameter values or introducing additional constraints on microbial physiology.

### 4.2.1 Correlations between parameter values

Correlations between parameter values could effectively alleviate the occurrence of unstable EPs by simultaneously changing parameter values that appear on both sides of the inequality given by Eqs. (25) or (26), thereby ensuring that these conditions are always fulfilled even as parameter values change. Some evidence proving this to be effective is provided by, e.g., Hararuk et al. (2015), who used the four-pool AWB model (Allison et al., 2010) (similar to the SDBE model) for predictions of global carbon stocks. They prescribed, e.g., the uptake and depolymerization rate coefficients and the respective half-saturation constants to positively correlate with temperature. Thus, with the same directional change in temperature, these parameter values change in opposite directions with respect to their threshold values for stability (Table 7) – i.e., with a decrease in temperature, $v_f^p$ decreases, moving closer to its threshold; however, simultaneously, $K_f^p$ also decreases, moving further away from its threshold. Consequently, for a wide range of parameter values, the conditions for stability could be fulfilled. Indeed, Hararuk et al. (2015) reported that they did not observe any unstable equilibria with maximum-likelihood parameters in their global study.

Beyond the qualitative assessment of parameter thresholds (Table 7), explicit analytical expressions of the necessary or sufficient conditions for stability (as for $Z_{m \times m}$ in Sect. S2.3 in the Supplement) could be used to quantitatively assess what parameter correlations are required to ensure stability of equilibria across reasonable parameter ranges. However, for other kinetic formulations besides $m \times m$, these terms might become difficult to trace analytically.

### 4.2.2 Eco-evolutionary constraints on microbial traits

The observation that stability thresholds for parameters shift as environmental conditions change can further be interpreted in the light of expected variations in microbial functional traits. While some kinetic rate parameters might be correlated due to thermodynamics (e.g., temperature response of rate parameters), correlations among other parameters, like the investment into growth or extracellular enzyme production, might rather emerge as outcomes of eco-evolutionary processes that select specific combinations of traits in a given environment (Abs et al., 2023, 2024). These combinations of traits would manifest themselves as microbial life history strategies under different environmental conditions (e.g., Malik et al., 2020). Following this logic, changing environmental conditions could constrain the space for microbial physiological adaptation because microbial traits would need to ensure stability. For example, very inefficient microbes (having a low $y_m$) could not establish a stable equilibrium under very unfavorable conditions (low OC input and/or high DOC leaching, Fig. 4b) unless other traits change simultaneously. This reflects a basic principle of ecology: organisms have to be adapted to the environment they inhabit, and such adaptation is expressed through sets of coordinated traits. Currently, this basic principle has not been integrated into microbial-explicit SOC models (but see Abs et al., 2022, for a recent attempt to address this challenge), which can lead to matching specific environmental conditions with a (modeled) microbial population that is not able to sustain itself under those conditions.

While our results are restricted to equilibrium conditions in the absence of external perturbations (e.g., fluctuating environmental conditions), we can speculate that trait coordination should also emerge in a fluctuating environment (for an example of plant trait coordination under stochastic soil moisture, see Bassiouni et al., 2023). In fact, changes in the microbial community with relevance for SOC cycling can happen on different timescales, from hours (stress response) and weeks (changes in community composition) to years and decades (mutations) (Abs et al., 2024). Environmental fluctuations occur at all these scales, and microbial communities are well equipped to adapt to these stochastically changing environments.

Integrating soil microbial ecological understanding into microbial-explicit SOC models could thus yield alternative mathematical descriptions or parameter relations that could

prevent mismatching between parameter values and environmental conditions and ultimately improve model applicability (Georgiou et al., 2017). Evidence regarding the importance of microbial ecology and evolution for SOC cycling is accumulating (Abs et al., 2022, 2023). For instance, microbes have been found to invest more into the production of extracellular enzymes in soils with lower SOC contents (Calabrese et al., 2022; Malik et al., 2019), and density-dependent microbial mortality, a concept derived from ecological considerations, can effectively alleviate oscillatory behavior (Georgiou et al., 2017).

Yet, revisiting the proposed stability criterion from an eco-evolutionary perspective prompts a paradox. In a model that obeys the proposed stability criterion (Eqs. 25 and 26), a reduction in enzyme production will always lead to an increase in the depolymerization rate – a strong disincentive for microbial investment into enzyme production. We can resolve this paradox by realizing that, for the SDB and SDBE models analyzed (and applied by, e.g., Hararuk et al., 2015, but also including the original model formulation by Allison et al., 2010), there is no process (such as competition with other (micro-)organisms or abiotic removal of (accessible) SOC through, e.g., leaching or occlusion) that competes with microbes for SOC. Consequently, the only alternative to SOC degradation is its accumulation. In turn, this allows for high depolymerization rates with minimal microbial investment into enzyme production. This outcome also emerges from a mathematical analysis of optimal substrate utilization – without losses of substrate due to abiotic processes or competition, decomposers do not have any reason to invest in resource acquisition (Manzoni et al., 2023). Thus, a rigorous eco-evolutionary optimization approach (that, for instance, aims to maximize microbial growth rate) cannot be readily implemented with the current model structure but would require an extension of the model to account for competition for SOC. Our analysis instead demonstrates how (given the model structure) microbes might adapt to environmental conditions not to maximize their fitness but to attain a stable population. Whether stability and evolutionary fitness maximization are convergent (i.e., an organism at an evolutionary fitness maximum would also establish a stable population) poses an interesting question for future research.

### 4.2.3 Comparing approaches to avoid instability

It is important to note that, while our analysis shows all three of the approaches listed in Sect. 4.2 to be feasible means to avoid instability, it gives no indication of which of the approaches should be preferred. As different objectives and research questions have differing requirements (e.g., whether the occurrence of instabilities should globally be avoided by removing the positive DOC feedback or whether it suffices to constrain the available parameter space), we cannot give a general recommendation. However, an important distinction between the different approaches is that, while the first two

(removal of an explicit DOC representation and the dependency of uptake on microbial biomass) are simplifications of the system that might require further justification for specific use cases (Wutzler and Reichstein, 2013), the third approach can add realism to the model by explicitly considering the interaction between environmental conditions and the microbial community. In line with Abs et al. (2022, 2023, 2024) and Georgiou et al. (2017), we want to highlight the potential of using such ecologically consistent mathematical descriptions to improve current model formulations. In other words, we cannot simply add a biotic component to models without acknowledging that this component has to be "adapted" (as species and communities are in the real world) to the environmental conditions it is exposed to.

### 4.3 Implications

#### 4.3.1 Mathematical insights into microbially mediated SOC cycling

Currently, the debate on the implications of different model formulations for microbially mediated SOC cycling is ongoing (He et al., 2024; Lennon et al., 2024; Tao et al., 2023, 2024a). Our contribution provides an additional perspective to this discussion by leveraging mathematical requirements for stability. Based on our analysis, we cannot draw any conclusions regarding which (if any) model version is most suitable to realistically represent microbially mediated SOC cycling at the large scale. Microbial-explicit models including a DOC pool (SDB and SDBE models) create a positive feedback loop that allows for instabilities to occur, but this finding by no means indicates a shortcoming of such models. In fact, the positive feedback loop is a direct consequence of our conceptual understanding of the microbially mediated, enzymatically driven degradation of organic carbon substrates in soils (Kuzyakov et al., 2000; Kuzyakov, 2010). At small spatial scales, oscillatory behavior can be realistic (Manzoni and Porporato, 2007); additionally, the collapse of a local microbial population is not difficult to imagine (the alternative – and, perhaps, more realistic – outcome that microbes turn dormant cannot be described by these models). However, at large spatial scales, these behaviors are not observed, indicating that SOC cycling is more stable at these scales (Wang et al., 2014; Georgiou et al., 2017). This can be seen as being indicative of a scaling problem – the same process cannot be described by the same mathematical formulations across scales (Chakrawal et al., 2020; Wilson and Gerber, 2021). The proposed stability criterion, as well as the outlined approaches to avoid instability, could, in turn, guide the development of upscaled model formulations – suitable upscaled kinetics might employ one of the proposed approaches to avoid instability or could be designed to obey the proposed (note: very conservative; see further discussion below) stability criteria.

### 4.3.2 Applicability of the proposed stability criterion

We could identify a sufficient and necessary condition for the stability of the SDB model (Eqs. A11–A13). However, the condition we found is difficult to interpret and apply. We thus proposed a stricter but simpler sufficient condition for stability (Eq. 25). By comparing the SDB and SDBE model, we proposed that a similar constraint ($Z_{i \times j} \geq 0$, Eq. 26) would also hold as a sufficient condition for the SDBE model despite the Routh–Hurwitz stability criterion being not fully tractable analytically for this model version. Numerical analyses confirmed that the proposed sufficient condition ensures stability of the SDBE model within the vast parameter space we explored. However, these sufficient conditions are very conservative and can exclude a substantial fraction of the physically meaningful and stable equilibrium points. Further, despite a clear correlation between $Z_{i \times j}$ and the damping coefficient $\zeta$, the stability condition does not give direct insights into the oscillation behavior. How useful the stricter sufficient and necessary conditions would be in constraining model parameters – as compared to the simpler sufficient conditions – might depend on the specific model applications. Despite the potential challenges in evaluating these conditions, they can still be useful to understand the processes or parameter interactions that cause unstable EPs to occur and can guide ecology-informed model developments.

### 4.3.3 Conditions leading to instability

Our numerical analysis of the SDBE model indicates that instability of equilibrium points becomes more likely with decreasing carbon inputs, increasing DOC leaching, and low process rates (Fig. 4, Table 7). All these conditions are most likely to be met in high-altitude and/or high-latitude environments. This is in line with Hararuk et al. (2015), who observed the strongest oscillations (longest time to dampen oscillations, indicative of diminishing real parts of the eigenvalues) in their calibrated four-pool model in these regions. Therefore, analytical steady states of microbial explicit SOC models applied in high-altitude and/or high-latitude environments could be unstable, and analytical steady-state solutions could thus not be used reliably for the initialization of simulation runs or prediction of SOC stocks.

## 5 Conclusions

Microbial-explicit SOC models aim to improve the representation of SOC dynamics by accounting for its biotic controls. At very small spatial and temporal scales, their oscillatory behavior and potential for instability can reflect relevant (micro-)ecosystem processes (Manzoni and Porporato, 2007). However, if applied at larger scales, such as in Earth system models, these properties can result in unrealistic simulation outcomes (Georgiou et al., 2017; Wang et al., 2014). Here, we analyzed what processes can lead to instability

in these models. By comparing the stability behavior of an archetypal microbial-explicit SOC model (the AWB model; Allison et al., 2010) with some reduced model versions and with the stability analysis of similar models in the literature, we found that instability can occur in models that assume a positive feedback between the resupply of a microbial growth substrate (i.e., DOC) and its abundance. We found that stability is (sufficiently) conditional on the balance between the sensitivity of the depolymerization rate to changes in extracellular enzymes vs. SOC concentration. Based on these analyses, we suggest that instability can be avoided by selecting specific (1) model structures, (2) kinetic formulations, and/or (3) parameter relations or values. While these approaches can differ vastly, an emerging common theme is that acknowledging ecological principles and processes can be leveraged to improve model applicability. These findings have implications for further development of microbial-explicit models and potential upscaling approaches, calling for ecologically consistent model formulations and rigorous mathematical analysis of newly introduced models.

## Appendix A: Details on stability analyses

### A1 SBE model

The Jacobian matrix of the SBE model is given by TS3

$$
J_i^{\text{SBE}} = \begin{bmatrix} \dfrac{\partial \dot{S}}{\partial S} & \frac{\partial \dot{S}}{\partial B} & \frac{\partial \dot{S}}{\partial E} \\ \frac{\partial \dot{B}}{\partial S} & \frac{\partial \dot{B}}{\partial B} & \frac{\partial \dot{B}}{\partial E} \\ \frac{\partial \dot{E}}{\partial S} & \frac{\partial \dot{E}}{\partial B} & \frac{\partial \dot{E}}{\partial E} \end{bmatrix}
$$
$$
= \begin{bmatrix} -l_s - \dfrac{\partial P_i}{\partial S} & r_B d_B & d_E - \frac{\partial P_i}{\partial E} \\ y_B \frac{\partial P_i}{\partial S} & -(d_B + m_E) & y_B \frac{\partial P_i}{\partial E} \\ (y_m - y_B)\frac{\partial P_i}{\partial S} & m_E & (y_m - y_B)\frac{\partial P_i}{\partial E} - (d_E + l_E) \end{bmatrix}, \quad \text{(A1)}
$$

where $\frac{\partial x}{\partial y}$ is the partial derivative of $x$ with respect to $y$, and $\dot{x}$ is used to denote $\frac{\mathrm{d}x}{\mathrm{d}t}$ so that, e.g., $\frac{\partial \dot{S}}{\partial S} = \frac{\partial}{\partial S}\left(\frac{\mathrm{d}S}{\mathrm{d}t}\right)$.

### A1.1 Stability analysis of the abiotic equilibrium

First, we evaluate the parameter space in which the abiotic equilibrium is stable. Substituting the steady-state solutions for $Q_0$ given in Table 5 into $J_i^{\text{SBE}}$ (Eq. A1) yields

$$
J_i^{\text{SBE}}\big|_{*,0} = \begin{bmatrix} -l_s & r_B d_B & d_E - \frac{\partial P_i}{\partial E}\big|_{*,0} \\ 0 & -\beta & y_B \frac{\partial P_i}{\partial E}\big|_{*,0} \\ 0 & m_E & (y_m - y_B)\frac{\partial P_i}{\partial E}\big|_{*,0} - \alpha \end{bmatrix}. \quad \text{(A2)}
$$

For $Q_0$ to be stable according to the Routh–Hurwitz criterion, it is required that all the coefficients $a_i$ of the characteristic polynomial of $J_i^{\text{SBE}}\big|_{*,0}$ and, additionally, $a_1 a_2 - a_3$ be positive. By these means, we find that the stability of $Q_0$ is

conditional on the following sufficient and necessary condition (Sect. S2.1 in the Supplement):

$$\left.\frac{\partial P_i}{\partial E}\right|_{*,0} < \frac{\alpha\beta}{\eta}. \tag{A3}$$

### A1.2 Stability analysis of the biotic equilibrium

Next, we analyze the stability of the biotic equilibrium by evaluating the Jacobian matrix $J_i^{\mathrm{SBE}}$ (Eq. A1) according to its biotic steady states $Q_{1,i}$ (Table 5):

$$J_i^{\mathrm{SBE}}\big|_{*,1} = \begin{bmatrix} -l_s - \frac{\partial P_i}{\partial S}\big|_{*,1} & r_B d_B & d_E - \frac{\partial P_i}{\partial E}\big|_{*,1} \\ y_B \frac{\partial P_i}{\partial S}\big|_{*,1} & -\beta & y_B \frac{\partial P_i}{\partial E}\big|_{*,1} \\ (y_m - y_B)\frac{\partial P_i}{\partial S}\big|_{*,1} & m_E & (y_m - y_B)\frac{\partial P_i}{\partial E}\big|_{*,1} - \alpha \end{bmatrix}. \tag{A4}$$

To evaluate the Routh–Hurwitz criterion, it is convenient to re-express $P_i$ in terms of $\frac{\partial P_i}{\partial E}$ as follows:

$$P_i = \frac{\partial P_i}{\partial E} x_i^{-1} E, \tag{A5}$$

where the factor $x_i$ is introduced to maintain the generality of this substitution for any $P_i$, as defined in Table 2:

$$x_i = \begin{cases} 1 & \text{if } (i = m, f) \\ 1 - \frac{E}{K_r^p + E} & \text{if } i = r \\ 1 - \frac{E}{K_e^p + S + E} & \text{if } i = e. \end{cases} \tag{A6}$$

$x_i$ has the convenient property $0 < x_i \le 1$. The ensuing derivations hold for all values of $x_i$ as long as $0 < x_i \le 1$ and also for other kinetics $P_j$ not explored here, for which a $0 < x_j \le 1$ can be found in order to satisfy Eq. (A5).

Substituting Eq. (A5) and $B_{1,i}^* = \frac{\alpha y_B}{\eta} E_{1,i}^*$ (Table 5) into Eq. (12), evaluated at a steady state, yields

$$\frac{\mathrm{d}E_i}{\mathrm{d}t} = (y_m - y_B)\frac{\partial P_i}{\partial E}\Big|_{*,1} x_i^{-1}\big|_{*,1} E_{1,i}^* + m_E \frac{\alpha y_B}{\eta} E_{1,i}^* \\ - d_E E_{1,i}^* - l_E E_{1,i}^* = 0, \tag{A7}$$

from which, for $E_{1,i}^* \ne 0$, we find

$$\left.\frac{\partial P_i}{\partial E}\right|_{*,1} = \alpha x_i\big|_{*,1}\frac{1 - \frac{m_E y_B}{\eta}}{y_m - y_B} = \frac{\alpha\beta}{\eta} x_i\big|_{*,1}. \tag{A8}$$

With this definition, we obtain

$$J_i^{\mathrm{SBE}}\big|_{*,1} = \begin{bmatrix} -l_s - \frac{\partial P_i}{\partial S}\big|_{*,1} & r_B d_B & d_E - \frac{\alpha\beta}{\eta} x_i\big|_{*,1} \\ y_B \frac{\partial P_i}{\partial S}\big|_{*,1} & -\beta & y_B \frac{\alpha\beta}{\eta} x_i\big|_{*,1} \\ (y_m - y_B)\frac{\partial P_i}{\partial S}\big|_{*,1} & m_E & \alpha\left(x_i\big|_{*,1} - \frac{m_E y_B}{\eta} x_i\big|_{*,1} - 1\right) \end{bmatrix}. \tag{A9}$$

From this, it can be seen that the trace of $J_i^{\mathrm{SBE}}\big|_{*,1}$ (the sum of the diagonal entries) is always negative since $\frac{\partial P_i}{\partial S}\big|_{*,1} > 0$ and $x_i\big|_{*,1} \le 1$; thus, $a_1 > 0$. Likewise, it can be shown that all remaining coefficients of the characteristic polynomial are always positive and that $a_1 a_2 - a_3 > 0$ (see detailed analytical derivations in Sect. S2.1 in the Supplement). Thus, all physically meaningful biotic equilibrium points of the three-pool SBE model are stable.

### A2 SDB model

The Jacobian matrix $J_{i \times j}^{\mathrm{SDB}}$ of the SDB model evaluated around the biotic equilibrium is given by

$$J_{i \times j}^{\mathrm{SDB}}\big|_* = \begin{bmatrix} -\frac{\partial P_i^{\mathrm{qss}}}{\partial S}\big|_* & 0 & -\frac{\partial P_i^{\mathrm{qss}}}{\partial B}\big|_* + f_D r_B d_B \\ \frac{\partial P_i^{\mathrm{qss}}}{\partial S}\big|_* & -\frac{\partial U_j}{\partial D}\big|_* - l_D & \frac{\partial P_i^{\mathrm{qss}}}{\partial B}\big|_* - \frac{\partial U_j}{\partial B}\big|_* + (1 - f_D)r_B d_B + m_E \\ 0 & y_m \frac{\partial U_j}{\partial D}\big|_* & y_m \frac{\partial U_j}{\partial B}\big|_* - m_E - d_B \end{bmatrix}. \tag{A10}$$

Since only the biotic EP of the SDB model is analyzed, we dropped the subscript 1 here.

Evaluating the Routh–Hurwitz criterion for the SDB model with the chosen kinetic formulations (Tables 2 and 4) yields the sufficient and necessary condition for stability as follows:

$$a_1 a_2 - a_3 = X_{i \times j} + Y_{i \times j} > 0, \tag{A11}$$

with

$$X_{i \times j} = \frac{\partial U_j}{\partial D}\Big|_* \left\{ \left(\frac{\partial U_j}{\partial D}\Big|_* + \frac{\partial P_i^{\mathrm{qss}}}{\partial S}\Big|_*\right) \right. \\ \underbrace{\left(\frac{\partial P_i^{\mathrm{qss}}}{\partial S}\Big|_* + y_m f_D r_B d_B - y_m \frac{\partial P_i^{\mathrm{qss}}}{\partial B}\Big|_*\right)}_{\text{Term giving rise to the sufficient condition (Eq. 25)}} \\ \left. + \pi \frac{\partial U_j}{\partial D}\Big|_* \right\} \tag{A12}$$

and

$$Y_{i \times j} = -\frac{\partial \dot{B}}{\partial B}\Big|_{*,j} \left\{ \frac{\partial P_i^{\mathrm{qss}}}{\partial S}\Big|_* \left(\frac{\partial P_i^{\mathrm{qss}}}{\partial S}\Big|_* - \frac{\partial \dot{B}}{\partial B}\Big|_{*,j}\right) \right. \\ \left. + \frac{\partial U_j}{\partial D}\Big|_* \left(2 \cdot \frac{\partial P_i^{\mathrm{qss}}}{\partial S}\Big|_* + y_m f_D r_B d_B - y_m \frac{\partial P_i^{\mathrm{qss}}}{\partial B}\Big|_* + \pi\right) \right\}, \tag{A13}$$

both shown for $l_D = 0$ for conciseness (see full expressions with $l_D \ge 0$ and detailed analysis of all coefficients of the characteristic polynomial in Sect. S2.2 in the Supplement). The sufficient condition for stability given by Eq. (25) in the main text holds for any $l_D \ge 0$. Beyond this sufficient condition for stability, the additional positive term $\pi \frac{\partial U_j}{\partial D}\big|_*$ in Eq. (A12) might indicate the stabilizing influence of the balancing loop $B_2$ (Fig. 2b) and the recycling of ENZ and MBC to SOC (compare Eq. 24).

$\frac{\partial \dot{B}}{\partial B}\big|_{*,j}$ in Eq. (A13) is the lower-right entry of $J_{i \times j}^{\mathrm{SDB}}\big|_*$ (Eq. A10) and is given by

$$\left.\frac{\partial \dot{B}}{\partial B}\right|_{*,j} = \frac{\partial}{\partial B}\frac{\mathrm{d}B}{\mathrm{d}t}\Big|_{*,j} = y_m \frac{\partial U_j}{\partial B}\Big|_* - m_E - d_B. \tag{A14}$$

For any choice of $U_j$ that is linear in $B$ (as is the case for $U_m$ and $U_f$ – compare Table 2), we find from solving $\frac{\mathrm{d}B}{\mathrm{d}t}$ (Eq. 16) at a steady state that $y_m \frac{\partial U_{(m,f)}}{\partial B}\big|_* = m_E + d_B$; thus

$\frac{\partial \dot{B}}{\partial B}\big|_{*,(m,f)} = 0$ so that $Y_{i\times(m,f)} = 0$ (Eq. A13). The only necessary and sufficient condition for the stability of the SDB model in these cases is thus $X_{i\times j} > 0$ (Eq. A12) (for $l_D = 0$; see Sect. S2.2 in the Supplement for the corresponding necessary condition for $l_D > 0$).

For linear uptake kinetics $U_1$, that is

$$U_1 = v_l^u D, \tag{A15}$$

$Y_{i\times l}$ (Eq. A13) does not vanish from Eq. (A11) (since $\frac{\partial U_1}{\partial B}\big|_* = 0$) or, consequently, from Eq. (A14). CE5

$$-\frac{\partial \dot{B}}{\partial B}\bigg|_{*,l} = m_E + d_B > 0 \tag{A16}$$

Because of its additional positive components, $Y_{i\times j}$ can be positive even if the sufficient condition of Eq. (25) is not fulfilled. Thus, using $U_1$ can help to ensure the positivity of all the coefficients of the characteristic polynomial and $a_1 a_2 - a_3 > 0$.

## A3  SDBE model

The Jacobian matrix of the four-pool SDBE model with $L_S = L_E = 0$ and $y_B = y_m$ (Eq. 6) (Table 4) and evaluated around the biotic equilibrium is given by

$$J_{i\times j}^{\text{SDBE}}\big|_* = \begin{bmatrix} -\frac{\partial P_i}{\partial S}\big|_* & 0 & f_D r_B d_B & -\frac{\partial P_i}{\partial E}\big|_* \\ \frac{\partial P_i}{\partial S}\big|_* & -l_D - \frac{\partial U_j}{\partial D}\big|_* & (1-f_D)r_B d_B - \frac{\partial U_j}{\partial B}\big|_* & d_E + \frac{\partial P_i}{\partial E}\big|_* \\ 0 & y_m \frac{\partial U_j}{\partial D}\big|_* & y_m \frac{\partial U_j}{\partial B}\big|_* - (m_E + d_B) & 0 \\ 0 & 0 & m_E & -d_E \end{bmatrix}. \tag{A17}$$

Further details on the stability analysis of the SDBE model are given in Sect. S2.3 in the Supplement.

*Code availability.* The code used for the numerical stability analysis and comparison with the proposed sufficient stability condition is publicly available at Zenodo https://doi.org/10.5281/zenodo.12749207 (Schwarz, 2024).

*Data availability.* All data used in this paper are available in the Supplement below. TS4

*Supplement.* The supplement related to this article is available online at: https://doi.org/10.5194/bg-21-1-2024-supplement. TS5

*Author contributions.* ES and SM conceptualized the study. ES and SG led the study and SM and SB assisted the formal analysis and investigation. All the authors discussed the results together. ES wrote the original draft of the paper and produced the figures, with feedback from SM. All authors reviewed and commented on the original draft of the paper and its revisions. CE6

*Competing interests.* The contact author has declared that none of the authors has any competing interests.

ther geographical representation in this paper. While Copernicus Publications makes every effort to include appropriate place names, the final responsibility lies with the authors.

*Acknowledgements.* We thank Björn Lindahl for the inspiring discussions and valuable comments on the paper and two anonymous reviewers for their insightful comments and reflections.

*Financial support.* This research has been supported by the European Research Council (ERC) under the European Union's Horizon 2020 Research and Innovation Programme (grant no. 101001608), the Vetenskapsrådet (grant no. 2020-03910), and the Svenska Forskningsrådet FORMAS (grant no. 2021-02121).

The publication of this article was funded by the Swedish Research Council, Forte, FORMAS, and Vinnova. TS6

*Review statement.* This paper was edited by David Medvigy and reviewed by two anonymous referees.

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

**Remarks from the language copy-editor**

**CE1**  Please note that, due to a significant number of changes requested, those which are purely stylistic in nature and/or refer to adjustments to your own writing rather than the work of the copy-editor have not been inserted as only technical and necessary changes are allowed at this stage in order to avoid accidentally implementing errors. In case a request was unclear but possibly significant, a comment was left asking for clarification. Noted oversights have been corrected. Please see our proofreading guidelines at https://www.publications.copernicus.org/for_authors/proofreading_guidelines.html and, in particular, note that any further substantial changes will require approval from the handing editor. With this in mind, please provide an explanation for why the remaining changes you have requested should be inserted. Thank you.

**Remarks from the typesetter**

**TS1**  Please note that vector graphics (*.eps, *.ps) cannot be included in the PDFLaTeX for technical reasons. In addition, *.pdf figures cannot be included in the PDFLaTeX since certain fonts or other content cannot be embedded and such content would then not show up in some browsers or *.pdf viewers. As a result, affected figures might appear incomplete to some readers. Therefore, we only include *.png and *.jpg figures in the article *.pdf. However, since we also publish all articles in full-text HTML, we will provide your vector graphics as high-resolution figures so that readers are able to download and enlarge the figures for re-use (see e.g. https://www.hydrol-earth-syst-sci.net/23/1163/2019/). Please note that this high-resolution download is only possible if your figure has the Creative Commons Attribution 4.0 License (CC BY) applied. This is the case for the figures compiled by you or your co-authors. If you cite a figure from another paper that is not distributed under the Creative Commons Attribution License, the figure is identified as protected and the download link will be hidden. Thank you very much for your understanding.