# Peer review of "When and why microbial-explicit soil organic carbon models can be unstable"

_EGUsphere, 2024_

## Author Response (AR1)

*We very much appreciate the supportive, interested and helpful comments by two anonymous reviewers. We addressed all reviewer comments as suggested in our initial public replies. We made further minor changes aiming to further increase clarity of our manuscript (listed after the replies to reviewer comments).*

*All changes are highlighted in the PDF version of the revised manuscript.*

*We address each individual reviewer comment below and indicate the changes we have made to the manuscript in response. The original reviewer's comments are stated in regular typeface, our initial responses in black italic and our final changes to the manuscript in red italic.*

**Comments by Reviewer #1**

*We thank anonymous reviewer #1 for their very helpful and relevant comments.*

Schwarz and collaborators used analytical, numerical and descriptive tools to investigate when and why unstable equilibrium points occur in an archetypal four-pool microbial-explicit model and some simplified versions of it. This study has obtained sufficiently (and necessary) conditions for the stability of the equilibrium point (EPs) of different model version through rigorous mathematical derivations and numerical simulations, which is useful to understand the process or parameter interactions that cause unstable EPs to occur and to guide ecology-informed model developments.

The results of this study point to three strategies to avoid unstable EPs in microbial-explicit SOC models. Besides these mathematical implications, the authors should focus more on the microbial ecological rationale rather than simply forcing the theoretical mathematical model stable. For example, in the real world, whether there exists (1) a positive feedback coupling between microbial growth substrate available for uptake and its resupply? and (2) the dependency of uptake on microbial biomass? What if these two phenomena are indeed facts?

> *Our study was motivated by the need for more diversified descriptions of soil organic carbon dynamics in Earth-system models (ESMs), and the ensuing proposition of using microbial-explicit models in ESMs. To meet this end, it is relevant to understand the mathematical properties of these models and potential problems. We thus focused our mathematical analysis on a suite of archetypal microbial-explicit model formulations that are being tested for implementation in ESMs. We found that for these microbial-explicit SOC models whether or not the temporal dynamics of a labile substrate (dissolved organic carbon, DOC) pool are explicitly represented has important consequences for a models' mathematical properties – because of the positive feedback they create. Whether these models are best suited to realistically represent SOC dynamics was not the aim of our study. A debate on suitable structures of microbial-explicit models is currently ongoing (He et al. (2024) and replies – as well as a recent opinion piece by Lennon et al. (2024)). Our study adds another aspect to this important discussion. It might also be important to recall that a practical reason why we need to understand instabilities is that analytically determined steady state values can be used for model initialization. If such equilibria are unstable, the model will exhibit erratic (and probably unrealistic) behaviour. Thus, our work is important not only for understanding of ecological contexts that might lead to collapse of soil functions (probably at the micro-scale, as discussed below), but also for avoiding instabilities in models applying nonlinear equations at large scales.*

*Within this context, we acknowledge the importance of the raised questions. A short answer would be that both (1) the feedback mechanism and (2) the microbial biomass dependence of uptake are rooted in ecological theory (or at least are based on our ecological understanding). We explain in more details as follows:*

1) *Generally, for conceptualizing substrate-microbe interaction in soil, the relevance of depolymerization of polymeric substrates through extracellular enzymes is well established in soil ecology. If we consider that these extracellular enzymes are produced predominantly by microbes and they make previously inaccessible substrate available for microbial consumption, this generates the basis for the positive feedback loop represented by the analysed models. Therefore, this feedback loop is a direct consequence of our conceptual understanding of microbial resource acquisition. The priming effect might be regarded as an empirical example of this positive feedback: the temporary provision of labile substrate allows microbes to produce enzymes and break down substrate that was previously not degraded because of energetic limitations (Kuzyakov, 2010; Kuzyakov et al., 2000). However, several factors can influence the strength of such a positive feedback. Importantly, not all inputs of labile carbon are due to microbial depolymerization, such as root exudation, desorption or leaching from litter. The relative importance of these inputs can vary locally (e.g. between the rhizosphere and the bulk soil). If these inputs represent a considerable and constant carbon flux, the positive feedback coupling is partly broken – i.e. the positive feedback exists on a continuum from strong feedback when enzymatic reactions contribute the most to DOC formation to weak feedback when inputs independent of microbial activity are dominant. In the analysed models this continuum is described through the factor $f_I$, which prescribes how the organic carbon input is partitioned between SOC and DOC. As expected, in the extreme case of all input going to the labile DOC pool, we did also not observe instability any longer (p. 24, line 459-461, and Supplementary Information (SI) Section 2.3 and Fig. S5).*

2) *Mechanistically we could argue that without microbial biomass, there would be no uptake (we would arrive at the "abiotic" EP), and further that the larger the (active) microbial community/biomass, the larger the uptake flux. However, whether this dependency of uptake on microbial biomass is first-order, as assumed by multiplicative and forward Michaelis-Menten kinetics (fMM) can be debated – e.g. density effects could limit the per-biomass uptake rate as the microbial community grows; and importantly only the active fraction of the microbial community contributes to metabolic processes. These phenomena might (implicitly) be better captured by reverse Michaelis-Menten (rMM) or the equilibrium chemistry approximation (ECA) kinetics. However, Tang & Riley (2019) argued that fMM might be a valid approximation of microbial uptake kinetics – though this might not hold in carbon-rich organic soils where microbial biomass can be larger than in carbon-poor mineral soils.*

*Based on our findings, removing the positive feedback coupling between abundance and resupply of the growth substrate (by removing an explicit representation of DOC) or removing the dependence of microbial uptake on microbial biomass can help to avoid the occurrence of unstable equilibrium points in microbial-explicit models. However, with this we do not mean that these model formulations are to be preferred above others that for instance have an explicit representation of DOC. In fact, we state e.g. that models neglecting an explicit representation of DOC "might have shortcomings in cases where DOC dynamics become important e.g. if drying-rewetting dynamics or leaching are relevant" (p. 24, lines*

*470-471), and that the choice of how unstable EP's are avoided depends on the research question at hand and highlight the potential of the third approach—i.e., adapting parameter values in an ecologically consistent way (p 25-26, lines 523-528).*

*Following the reviewers' suggestion to "focus more on the microbial ecological rationale rather than simply forcing the theoretical mathematical model stable" we will extend the discussion based on the above reply and by highlighting that while the first two approaches are based on simplifications of the modelled system that might require further justification, the third approach in fact aims to add realism to microbial-explicit models, as it proposes to better acknowledge microbial ecology.*

*We added a new subsubsection "4.2.3 Comparing approaches to avoid instability" (lines 506-518). Here we expand our previous argument (lines 520-528 in the old manuscript) based on the above discussion. We further pick up the above discussion points in the new subsubsection "4.3.1 Mathematical insights on microbial mediated SOC cycling" (lines 520-536) in combination with points raised by reviewer 2. We now also explicitly mention the distinction between the different approaches in the abstract (lines 11-15).*

Regarding the third strategy, i.e., choosing parameter values to meet the sufficient and/or necessary conditions for stability, one may argue this could be feasible for a theoretical model with time-invariant carbon input rates and parameters. However, for the real field conditions, both the carbon input rates and parameters vary with time due to the biotic and abiotic effects. It could happen that the parameter values meet the so-called stability conditions sometimes but disobey the conditions at other times.

*Spatial and temporal variability are undoubtedly important controls on ecosystem dynamics and functioning. Locally, on small spatial and temporal scales it is not difficult to imagine that environmental fluctuations can lead to diverse behaviours including also the collapse of local microbial populations. However, at large spatial scales this is not observed. E.g. the litter removal data compiled by Georgiou et al. (2017) did not show an extinction of microbes even after decades without inputs. That being said, microbes in model simulations might not directly go extinct if parameter values would temporarily lead to unstable EP's but could recover if parameter values change back fast enough (though once the "abiotic" state is reached, recovery is impossible). Yet, determining stochastic steady states and their stability is beyond the scope of our study.*

*Adding to the complexity, not only do carbon input rates and rate parameters fluctuate but so does the microbial community and its functions. Microbial community adaptation to changing environmental condition happens at different timescales ranging from hours (stress response), and weeks (changes in community composition), to years and decades (evolutionary changes) (Abs et al., 2024). These considerations are still largely missing in microbial-explicit models trailed for application in ESM. The third strategy we lined out proposes to capture some of these processes implicitly by introducing appropriate relationships between parameters encoding microbial traits with ecological relevance and environmental conditions. We argue that allowing for adaptation of the modelled microbial community to different environmental conditions by varying microbial trait parameters as a function of environmental conditions could be an effective tool to avoid instability in these models while at the same time making them more consistent with ecological theory.*

*We added an additional paragraph based on the above arguments to the discussion under section 4.2.2 (lines 478-483).*

Therefore, I would like to see additional discussion on the limitation of applying theoretical analyses to real ecosystem, particularly regarding the stability analyses.

*Thank you for these helpful questions and reflections. We will expand the discussion with the above outlined arguments.*

*We addressed the raised points and discussion with several extensions of the discussion. Please see detailed answers above.*

Minor comments:

Page 3, Line 36-39 (Fig. 1 caption): insert a comma between "dissolved organic carbon (DOC)" and "microbial biomass carbon (MBC)"

*Done.*

*Done.*

Page 5, Line 106: Table 4 or Table 2?

*We meant to point to the summary of analysed scenarios but this might not be appropriate at this point. We moved the reference to Table 4 to the end of the previous sentence.*

*"(summary of analysed scenarios in Table 4)" was added to line 106-107 and "(Table 4)" removed from line 110.*

Page 8, Section 2.3.2 Specify the time step and period of numerical simulations

*We numerically computed steady state values and eigenvalues of the Jacobian matrix by substituting numerical values in the analytical solutions. We did not run transient simulations requiring ODE solvers.*

Page 11, Line 227: Clarify "If abiotic loss of SOC and are neglected"

*Changed to "If abiotic loss of SOC is neglected".*

*Done (line 226).*

Page 14, Line 291: Change "larger then" to "larger than"

*Done.*

*Done*

Page 17, Caption of Figure 2: "four-pool SDB", while "three-pool SDB" in the main text

*We changed the first sentence in the caption to read "Simplified causal loop diagrams of the SBE (a) and SDB and SDBE models (b)." to avoid confusion.*

*Done*

Page 21, Table 7: briefly explain the meanings of marking the lower or upper threshold.

*We will add the below schematics to Table 7 to clarify the meaning of the thresholds, and further explain it in the caption.*

*We slightly modified our initial suggestion for a schematic (see below). We added the sentence "The meaning of the different thresholds is illustrated in the schematics on the right." to the table caption.*

[Figure]

**Comments by Reviewer #2**

*We thank anonymous reviewer #2 for their supportive comments and stimulating thoughts.*

**General Comments**

Schwartz et al. present a thorough and robust stability analysis of microbial-explicit biogeochemical models, showing how model structure, kinetics, and parameter space can create unstable equilibria. The phenomenon of instability in microbial models has been mentioned in passing in existing literature (e.g. Schimel and Weintraub 2003, Georgiou et al., 2017) but the analysis presented in this manuscript represents the most thorough investigation of this phenomenon to date. This makes the manuscript novel and interesting to the community of people interested in developing microbial-explicit soil carbon models. I have two main areas of feedback to improve the manuscript.

First, I would like the authors to make sure that this manuscript is accessible to the community of researchers who would benefit from understanding its main conclusions. To this end, I count 46 numbered equations in the manuscript in addition to those contained in tables, which is really quite a lot. I appreciate that these equations build support for the main findings of the paper, but I suspect that very few people will read this paper and work through all these equations. There is a significantly broader audience that will benefit from understanding the outcome of this analysis without working through all the math. With this in mind, I suggest the authors take steps to ensure that the main findings are easy to locate and understand for this audience. Although the supplement is quite large already, moving some of the less crucial equations to the supplement could improve readability. In the figures, finding ways to represent data in less abstract ways and providing conceptual interpretations will make the conclusions more actionable for readers of the manuscript. Specific comments below.

> *Thank you very much for this helpful suggestion. We agree that accessibility of the paper and its conclusions could be broadened. We suggest to move less crucial equations currently contained in the main text to the Appendix (appearing at the end of the main text) rather than the supplemental information, where they might be easier to locate for interested readers. Specifically, we suggest to move the Jacobian matrixes and technical details on the derivation of stability criteria. Additionally, we will extend Table 3 to include parameter groups (explaining their meaning were appropriate – see specific comment below).*

> *We moved all Jacobian matrices to the newly created Appendix section (appearing at the end of the manuscript before the Acknowledgements). We further incorporated the definition of the parameter groups (eq. 23-26 in the old manuscript) into Table 3 and moved most equations in Section 3.1.2 to the Appendix Section A1.2 (eq. 28, 30-35 in the old manuscript). Likewise, detailed equations in 3.2.2 where moved to Appendix Section A2 (eq. 39-42 & 44-45 in the old manuscript). All equation references were updated accordingly in the main text and the Supplementary Information. Moving equations to the Appendix required slight reformulations of the main text which is highlighted in the track changes version of the revised manuscript.*

My second main comment is that the paper could benefit from some discussion of stability as a realistic ecological/biogeochemical phenomenon. We should expect microbial-explicit models to produce realistic predictions at the scale of the mechanisms that they represent. Generally, the mechanisms in these models can be described over very small spatial scales where conditions can be assumed to be homogenous. Directly upscaling non-linear uptake and depolymerization kinetics in heterogenous environments does not preserve model behavior (Chakrawal et al., 2020). It is

probably worth some discussion then whether equilibrium is a realistic way to represent microbial dynamics in soil at all. Microbial populations in reality may very well oscillate and exhibit instability at the scale of mechanisms represented in microbial models, while still producing stable emergent behavior at larger scales.

> *We agree that the behaviors these models produce – including oscillations and instability – can very well be realistic at the microscale (i.e. the scales of the processes they describe). We touch on this in lines 35-39 and 551-554. Still, if these models are without further modifications used to describe processes at larger spatial scales (as commonly done), these behaviors are no longer realistic. I.e. models used to simulate SOC fate at the large scale must have stable equilibrium points to prevent erratic simulation outcomes (see also response to reviewer 1). We agree that this discrepancy might be indicative of a scaling problem. If non-linear microbial-explicit models accurately describe processes at the small scale – at which oscillatory and unstable behavior might occur – we need to adapt these formulations before applying them at larger spatial scales where we do not observe these behaviors. In this light, the approaches to avoid instability could inform development of suitable upscaling approaches. I.e. suitable upscaling approaches might lead to formulations that simplify the non-linear dynamics (such as approaches 1 & 2 in section 4.2) or that acknowledge spatial (and temporal) heterogeneity and ecology of microbial communities (approach 3).*

> *Regarding the scaling relations established by Chakrawal et al. (2020) we note that it can be a very interesting exercise to plug their upscaled kinetic descriptions into our stability criterion (substituting any of their upscaled kinetic descriptions to represent the depolymerization rate $P$ and then taking the respective partial derivatives). However, the lack of a closure of the equations presented by Chakrawal et al. (2020) limits meaningful insights from this exercise – i.e. we would need to know how the variance and covariance terms of their equations scale with state variables (as terms such as $\frac{\partial}{\partial s} cov(S, E)$ would need to be evaluated - $cov(S, E)$ being the spatial covariance between SOC ($S$) and extracellular enzyme ($E$) concentration). Considering such scaling relations as a potential solution to avoid instability could be regarded as a special case of adapting kinetic formulations to avoid instability. Though, as it does not directly emerge from our analysis we refrain from listing it in line with the other approaches we suggest in section 4.2. Still we agree that this is an important notion and we suggest to extend the discussion to acknowledge it.*

> *We added a new subsubsection "4.3.1 Mathematical insights on microbial mediated SOC cycling" (lines 520-536) were we address the above points together with points raised by reviewer 1. We further added a sentence to the conclusions addressing the implications of our results for potential upscaling approaches (lines 570-572).*

**Specific Comments**

218-222: This is one place where grouping parameters helps express the equilibrium solutions in a concise way, but it becomes very difficult for me to interpret any of these equations with the added layers of abstraction. When the authors then mention ω > 0 or , the significance of this fact is not clear because the expression is hidden behind a layer of abstraction. Is there any conceptual interpretation that can be added to help readers understand? Similar issue lines 285-290.8

> *The grouping parameters were primarily introduced to express steady states in a concise way. These groups don't necessarily have a specific conceptual meaning. We state that $\omega >$*

*0 not to give an interpretation of ω, but only to understand under what conditions steady states are positive (and thus physically meaningful). On the other hand, e.g. α and β can be identified as "enzyme turnover" and "microbial biomass turnover", respectively. We suggest to add meaningful descriptions to parameter groups were appropriate and to further move the definitions of these parameter groups to Table 3 in order to not break the text flow unnecessarily.*

*Done as suggested, see details in above answer.*

Table 5 and 6: What do the vertical bars in the B and E columns signify?

*Vertical bars were supposed to indicate that the expressions for $B^*_{k,i}$ and $E^*_{k,i}$ apply to all used kinetic formulations. We suggest to use curly brackets to indicate this more clearly. E.g. for Table 5:*

**Table 5.** Summary of steady state solutions of the three-pool $SBE$ model for different kinetics of depolymerization. The "biotic" equilibrium solutions for microbial biomass and extracellular enzymes have the same form for any chosen kinetic.

| | Kinetic ($i$) | $S^*_{k,i}$ | $B^*_{k,i}$ | $E^*_{k,i}$ |
|---|---|---|---|---|
| abiotic ($k=0$) | $i$ | $\frac{I}{l_S}$ | 0 | 0 |
| biotic ($k=1$) | $m$ | $\frac{\alpha\beta}{v^p_m\eta}$ | | |
| | $f$ | $\frac{\alpha\beta}{v^p_f\eta}\frac{K^p_f}{1-\frac{\alpha\beta}{v^p_f\eta}}$ | $\frac{\alpha y_B}{\omega}l_S\left(S^*_0-S^*_{1,i}\right)$ | $\frac{\eta}{\alpha y_B}B^*_{1,i}$ |
| | $r$ | $\frac{\alpha\beta}{v^p_r\eta}\frac{K^p_r\omega+I\eta}{\omega+l_S\frac{\alpha\beta}{v^p_r}}$ | | |
| | $e$ | $\frac{\alpha\beta}{v^p_e\eta}\frac{K^p_e\omega+I\eta}{\omega-\omega\frac{\alpha\beta}{v^p_e\eta}+l_S\frac{\alpha\beta}{v^p_e}}$ | | |

*Changed as suggested (Table 5 & 6).*

339-340: This feedback is interesting because it makes mathematical sense but I'm not sure if it makes sense in an eco-evolutionary view. Doesn't this imply that microbes could decrease constitutive or inducible enzyme production rates without losing access to SOC? If depolymerization is substrate-limited at equilibrium, then isn't producing extracellular enzymes a losing strategy from an individual fitness perspective? Discussion of this potential paradox could strengthen the paper and help identify future lines of research.

*This is a very interesting notion and points to a very interesting paradox – i.e. that in fact "producing extracellular enzymes [is] a losing strategy from an individual fitness perspective". Importantly, the model as it is currently formulated and applied (e.g. by Hararuk et al. (2015) but also in the original model described by Allison et al. (2010)) does not consider any competition for SOC substrate. Thus, the only alternative to enzymatic degradation of SOC is its accumulation. This means, in an eco-evolutionary perspective (e.g. if maximizing the growth rate of microbes), that microbes would reduce their enzyme investment to a minimal amount, SOC would accumulate dramatically, but microbes could still attain high growth rates, though at low per-biomass cost for enzyme production. The eco-evolutionary perspective only becomes relevant (and applicable) once there is a competing process that removes SOC. In theory this could either be competition for SOC by other (micro)organisms or abiotic processes that remove (accessible) SOC (erosion, leaching, occlusion, sorption, …). This outcome emerges also from a mathematical analysis of optimal substrate utilization—without losses of substrate due to abiotic processes or competition, decomposers do not have any reason to invest in resource acquisition (Manzoni et al., 2023).*

*Thus, an eco-evolutionary optimization approach might not be readily implemented with the current model structure but would require extension of the model to account for competition for SOC. Our analysis demonstrates instead how (given the model structure) microbes might adapt to environmental conditions not to maximize their fitness, but to attain a stable population (e.g. by adapting microbial parameters accordingly; 3rd approach described Section 4.2). Whether stability and fitness maximization are convergent (and e.g. an organism at a fitness maximum would also be able to establish a stable population) would warrant an own dedicated paper. We will include this important notion in our discussion. We note that Abs et al. (2022) in their pre-print are exploring the effect of eco-evolutionary dynamics on soil carbon fate by explicitly considering competition between microbes (using an adaptive dynamics approach).*

*We added an additional paragraph based on the above discussion to the discussion section 4.2.2 (lines 491-505) and changed the subsection name to "Eco-evolutionary constraints on microbial traits"*

Figure 3: There are a few ways that labeling and captioning on this figure could be clarified to help readers interpret the figure

- For figure 3a, I think it may be useful to label the axes with a conceptual description of the mathematical expression. If I understand correctly, I think we are seeing the sensitivity of depolymerization to changes in soil carbon on the x axis and sensitivity of depolymerization to changes in enzymes on the Y. Points below the line fit the conservative condition for stability, but points with damping values > 1 are stable (if oscillatory). Clearly indicating this in the figure labels, axes, and caption will help readers interpret this figure independently, even if they haven't worked through all equations. It may also be helpful to specify in the caption that the proposed condition isn't arbitrary and rather an extension of the criteria found in the simpler model.
- It is confusing that 3b and 3c have essentially the same axis but are shown on two graphs. Figure should include a color legend for stability instead of explaining in the caption. Alternatively, can the same color gradient be used in 3b and 3c that is in 3a?

*We suggest to update the figure according to the reviewer's comments. Panels b and c are different in that they represent steady states that fulfill (b) or do not fulfill (c) the stability criterion. Hence the x-axes either have positive (b) or negative values (c). We indicated this difference more clearly now. We replaced absolute numbers of stable and unstable points now by their relative occurrence (%-values in b and c).*

[Figure]

*We updated the figure as suggested and added the sentence "The proposed stability condition $Z_{f\times f} \geq 0$ (eq. 26) is derived from an analytical evaluation of the simpler SDB model." to the caption and updated the caption text to reflect all other changes to the figure.*

Figure 4: Color scheme for 4b-d would benefit from a legend on the graph rather than in-text description. It is also confusing that the color scheme is the same as 4a, but corresponds to a different variable. I would also suggest labeling axes with variable names instead of single letters (i.e. Biomass decay rate) so that readers don't have to refer back to the parameter table to interpret the figure.

*We suggest to update the figures according to the reviewer's comments. Instead of using a legend for the color code of lines in b-d we prefer to have the description directly at the lines. We now swapped panels b and c so that this description together with the legend describing the different line styles appears in the upper right corner where it might be more intuitively found by readers.*

*We updated the figure as suggested with minor edits (see below) and altered the caption text to reflect these changes.*

[Figure]

Figure 5: Please clarify – are grey points stable but not plausible?

*Correct, we suggest to adapt the caption text as following to make this clearer:*

*"Color-coded points are stable and plausible steady state solutions, the color code indicating the value of the damping coefficient. Grey points are stable but not plausible steady state solutions and black points are physically meaningful but unstable steady state solutions."*

*Changed as suggested, also in Supplemental Figures S3 and S4.*

465: This is good and helpful for readers. It would be helpful to add 1-2 sentences in the abstract that summarize these approaches to avoiding instability.

*We propose to change lines 11-13 and add to it (altered text is highlighted):*

*Principally, three distinct strategies can avoid instability: 1) negligence of explicit DOC dynamics, 2) biomass independent uptake rate or 3) co-variance between parameter values to obey the stability criterion. While the first two approaches simplify some mechanistic processes, the third approach points to the interactive effects of environmental conditions and parameters describing microbial physiology, highlighting the relevance of basic ecological principles for avoidance of unrealistic (i.e. unstable) simulation outcomes.*

*Changed as suggested (lines 11-15).*

515: This may be a good place to include some discussion about whether microbial equilibrium is , either as a scale issue, or an eco-evolutionary issue, as discussed in comments above.

*We suggest to expand the discussion on the above points either here or where it comes naturally considering all arguments.*

*We extended the discussion as explained in the detailed comments above.*

**Technical Corrections**

382: analytical analysis is redundant

*Done.*

*Done*

**Citation**

Chakrawal, A., Herrmann, A. M., Koestel, J., Jarsjö, J., Nunan, N., Kätterer, T., & Manzoni, S. (2020). Dynamic upscaling of decomposition kinetics for carbon cycling models. *Geoscientific Model Development*, *13*(3), 1399-1429.

**Additional changes to the manuscript**

*We corrected spelling mistakes and updated some formulations to increase clarity of the manuscript.*

*We slightly modified Figure 2 (see below) to better highlight the difference between model structures leading to instability and updated the color-scheme to be more accessible. We updated the caption text to reflect these changes and added a pointer to the Materials & Methods section to assist reading the causal loop diagrams.*

[Figure]

*We added a brief broader perspective at the start of the discussion (lines 385-388).*

*We added a relevant additional discussion point based on recently published literature (lines 437-440).*

*We changed the name of Section "4.3 Relevance for model applications" to "4.3 Implications" and introduced further sub-sections to improve structure.*

*We thank both reviewers for their valuable additions to the manuscript and acknowledge this in the Acknowledgements.*

*All newly introduced references were added to "References".*

**Additional changes to the Supplemental Information**

*We corrected spelling mistakes and updated some formulations to increase clarity of the manuscript.*

*References were updated according to journal requirements.*

**References used in replies**

Abs, E., Chase, A. B., Manzoni, S., Ciais, P., & Allison, S. D. (2024). Microbial evolution—An under-appreciated driver of soil carbon cycling. *Global Change Biology*, *30*(4), e17268. https://doi.org/10.1111/gcb.17268

Abs, E., Saleska, S., & Ferriere, R. (2022). *Microbial eco-evolutionary responses amplify global soil carbon loss with climate warming* [Preprint]. In Review. https://doi.org/10.21203/rs.3.rs-1984500/v1

Allison, S. D., Wallenstein, M. D., & Bradford, M. A. (2010). Soil-carbon response to warming dependent on microbial physiology. *Nature Geoscience*, *3*(5), 336–340. https://doi.org/10.1038/ngeo846

Chakrawal, A., Herrmann, A. M., Koestel, J., Jarsjö, J., Nunan, N., Kätterer, T., & Manzoni, S. (2020). Dynamic upscaling of decomposition kinetics for carbon cycling models. *Geosci. Model Dev.*, *13*(3), 1399–1429. https://doi.org/10.5194/gmd-13-1399-2020

Georgiou, K., Abramoff, R. Z., Harte, J., Riley, W. J., & Torn, M. S. (2017). Microbial community-level regulation explains soil carbon responses to long-term litter manipulations. *Nature Communications*, *8*(1), 1223. https://doi.org/10.1038/s41467-017-01116-z

Hararuk, O., Smith, M. J., & Luo, Y. (2015). Microbial models with data-driven parameters predict stronger soil carbon responses to climate change. *Global Change Biology*, *21*(6), 2439–2453. https://doi.org/10.1111/gcb.12827

He, X., Abramoff, R. Z., Abs, E., Georgiou, K., Zhang, H., & Goll, D. S. (2024). Model uncertainty obscures major driver of soil carbon. *Nature*, *627*(8002), E1–E3. https://doi.org/10.1038/s41586-023-06999-1

Kuzyakov, Y. (2010). Priming effects: Interactions between living and dead organic matter. *Soil Biology and Biochemistry*, *42*(9), 1363–1371. https://doi.org/10.1016/j.soilbio.2010.04.003

Kuzyakov, Y., Friedel, J. K., & Stahr, K. (2000). Review of mechanisms and quantification of priming effects. *Soil Biology and Biochemistry*, *32*(11–12), 1485–1498. https://doi.org/10.1016/S0038-0717(00)00084-5

Lennon, J. T., Abramoff, R. Z., Allison, S. D., Burckhardt, R. M., DeAngelis, K. M., Dunne, J. P., Frey, S. D., Friedlingstein, P., Hawkes, C. V., Hungate, B. A., Khurana, S., Kivlin, S. N., Levine, N. M., Manzoni, S., Martiny, A. C., Martiny, J. B. H., Nguyen, N. K., Rawat, M., Talmy, D., … Zakem, E. J. (2024). Priorities, opportunities, and challenges for integrating microorganisms into Earth system models for climate change prediction. *mBio*, e00455-24. https://doi.org/10.1128/mbio.00455-24

Manzoni, S., Chakrawal, A., & Ledder, G. (2023). Decomposition rate as an emergent property of optimal microbial foraging. *Frontiers in Ecology and Evolution*, *11*, 1094269. https://doi.org/10.3389/fevo.2023.1094269

Tang, J., & Riley, W. J. (2019). Competitor and substrate sizes and diffusion together define enzymatic depolymerization and microbial substrate uptake rates. *Soil Biology and Biochemistry*, *139*, 107624. https://doi.org/10.1016/j.soilbio.2019.107624